# The Early Cretaceous Granitoids and Microgranular Mafic Enclaves of Sanguliu Pluton, the Liaodong Peninsula: Implications for Magma Mixing and Decratonic Gold Mineralization in the Eastern North China Craton

Taotao Wu, Cong Chen *, Dongtao Li, Yan Zhao, Chunqiang Zhao and Yongheng Zhou

Shenyang Center of Geological Survey, China Geological Survey, Shenyang 110034, China
* Correspondence: chencong@mail.cgs.gov.cn

**Abstract:** Some Early Cretaceous granitoids characterized by abundant mafic microgranular enclaves (MMEs) formed by magma mixing have been associated with gold deposits in the eastern North China Craton (NCC). However, the genetic connection of magma mixing with gold mineralization remains unclear. The zircon U–Pb ages and in situ Lu-Hf isotopic compositions, whole-rock major- and trace-element and Sr–Nd–Pb isotopic compositions, as well as EPMA biotite compositions, were presented for the Sanguliu granodiorite and enclaves in the Liaodong Peninsula in order to obtain insights into the spatial and temporal distribution, and internal connection of magma mixing with the decratonic gold deposits in the eastern NCC. The Sanguliu granodiorite yielded coeval formation ages with the enclaves (~123 Ma), and their acicular apatites and plagioclase megacrysts suggest that the enclaves were formed by mixing between mafic and felsic magmas. Geochemically, the Sanguliu granodiorite is high-K calc-alkaline I-type granite, with an initial $^{87}Sr/^{86}Sr$ ratio of 0.70552 to 0.71470 and strongly negative $\varepsilon_{Nd}(t)$ (−11.4 to −21.3) and zircon in situ $\varepsilon_{Hf}(t)$ values (−15.1 to −25.4), indicating that the felsic magmas were ancient lower crust with the involvement of mantle-derived materials. Meanwhile, the enclaves have high MgO (4.18 to 6.17 wt.%), Cr (45.91 to 290.04 ppm), and Ni (19.65 to 88.18 ppm) contents, with high $Mg^{\#}$ values of 50 to 57 at intermediate $SiO_2$ contents (53.68 to 55.78 wt.%), highly negative $\varepsilon_{Nd}(t)$ values (mostly −18.42 to −22.03), and in situ zircon $\varepsilon_{Hf}(t)$ values (−18.6 to −22.7), indicating that the mafic magma was mainly derived from the enriched lithospheric mantle. Furthermore, the biotites from the Sanguliu granodiorite clustered between the MH and NNO buffers in the $Fe^{2+}$–$Fe^{3+}$–Mg diagram. This, combined with the high Ce/Ce* ratios (1.30 to 107.18) of the zircons, indicates that the primary magmas forming the Sanguliu granodiorite had a high oxygen fugacity, which is favorable for gold mineralization. These findings, together with previous studies of the Early Cretaceous granitoids with enclaves in the eastern NCC, suggest that magma mixing commonly occurred during 110–130 Ma and is temporally, spatially, and genetically related to decratonic gold systems in eastern NCC.

**Keywords:** mafic microgranular enclaves; magma mixing; high oxygen fugacity; enriched lithospheric mantle; decratonic gold deposit; eastern North China Craton

## 1. Introduction

Granites commonly contain lithic inclusions or bodies, generally referred to as enclaves, whose nature and origin are documented [1,2]. Considering their occurrences, microstructural features, and relationship with host granites, enclaves have been grouped into (1) xenoliths, as solid fragments of wall rock, (2) restite, as segregated refractory source materials left after a partial melting event, (3) autolith, as early crystallized cumulus phases or segregated mafic phases or fragments of a border rock series of a felsic pluton, (4) microgranular enclaves representing felsic, mafic, and mafic-felsic (intermediate) hybridized magmas entrained and mingled into felsic host magma, and (5) fragments of synplutonic

mafic dyke intrusions [2]. In general, mafic microgranular enclaves (MMEs) are indicators of magma mingling and mixing processes and provide critical information regarding magma chamber processes and dynamics [2–4].

The eastern North China Craton (NCC) is characterized by widespread Early Cretaceous granitic intrusions. Some of these granitic intrusions contain abundant MMEs, which are considered a result of magma mixing and mingling via the intrusion of mafic magmas into felsic magma chambers [5–11]. Previous studies have mainly focused on field observations, the petrological characteristics of the host granitoids and MMEs, and the geochemistry of the host granitoids [5,6,9]. However, further MME information is limited, especially regarding their formation ages, magma sources, and mechanisms. Moreover, only the Gudaoling intrusion-enclosing MMEs in the Liaodong Peninsula have been generally researched. The intrusion is considered to have formed through a complex, multi-stage hybridization process that involved mantle- and crustal-derived magmas and several concomitant magmatic processes [7,8].

Furthermore, the mixing and mingling processes involving crustal- and mantle-derived magma may contribute to the formation of endogenetic metal deposits, especially gold deposits [10,12,13]. The Early Cretaceous granitic plutons enclosing MMEs formed by magma mixing and mingling, such as the Guojialing and Weideshan granitic intrusions, may be associated with gold deposits in the Jiaodong Peninsula in the eastern NCC [10]. However, the role of magma mixing in gold mineralization remains unclear. New field observations have identified the occurrence of MMEs in the Sanguliu intrusion, which hosts the large-scale Wulong gold deposit in the Liaodong Peninsula. The petrogenesis of the Sanguliu intrusion has not yet been investigated. The genetic relationships between the Sanguliu intrusion and the Wulong gold deposit remain controversial. Li et al. proposed that the Sanguliu intrusion is a metallogenetic intrusion of the Wulong gold deposit [14]. However, Wei et al. believed that the Sanguliu intrusion did not directly provide any ore-forming fluids or materials for Wulong gold mineralization but had similar Sr and Nd isotopic compositions to the deeper source material of the Wulong deposit [15]. The existence of deeper magma as the metallogenic intrusion has also been supported by Yu et al. [16] and Chen et al. [17]. Therefore, research on the Sanguliu intrusion may provide critical information regarding the relationship between magma mixing and gold mineralization.

On the basis of the field observation and petrography, this paper presents new zircon U-Pb ages and in situ Lu-Hf isotopes, whole-rock major and trace elements and Sr-Nd-Pb isotopic data of the MMEs and host Sanguliu granodiorite. The mineral compositions of biotite from the Sanguliu granodiorite were also analyzed. Based on the results, the petrogenesis and magma source of the Sanguliu granodiorite and monzodioritic enclaves were determined. We also discuss the relationship between the Sanguliu intrusion and the Wulong gold deposit and propose a new petrogenetic-dynamic model for these rocks. Furthermore, we review the published data for other Early Cretaceous granitoids enclosing MMEs in the Jiaodong, Liaodong, and Taihang areas in order to constrain the temporal and spatial distribution and identify any genetic relation between decratonic gold deposits and magma mixing in the eastern NCC.

## 2. Regional Geology

The Liaodong Peninsula is located in the eastern part of the NCC at the western margin of the Pacific Plate (Figures 1 and 2a). The exposed rocks in the area comprise metamorphosed Archean-Paleoproterozoic basement, Mesoproterozoic-Paleozoic post-cratonization cover sequences, and a series of Mesozoic intrusive and volcanic rocks (Figure 2b). The metamorphosed basement sequences include the Archean supracrustal rocks and diorite–tonalite–granodiorite suites, and the Paleoproterozoic metamorphosed sedimentary and volcanic successions of the Liaohe Group, as well as associated granitic and mafic intrusions. The Paleoproterozoic sedimentary and volcanic successions of the Liaohe Group metamorphosed from greenschist to the lower amphibolite facies and locally up to the granulite facies [18], which are conventionally divided into five formations:

Liangzishan, Li'eryu, Gaojiayu, Dashiqiao, and Gaixian Formations [19]. The lowermost Liangzishan Formation is composed of basal conglomerate-bearing quartzites, transitioning upwards to chlorite-sericite quartz schists, phyllites, garnet-bearing mica schists, minor graphite-bearing garnet-staurolite mica schists, and kyanite-bearing mica schists. It is conformably overlain by the Li'eryu and Gaojiayu Formations, which consist of boron-bearing fine-grained felsic gneiss, amphibolites, and mica quartz schist. Overlying the Gaojiayu Formation is the Dashiqiao Formation, which is composed predominantly of dolomitic marbles intercalated with minor carbonaceous slates and mica schists. The uppermost Gaixian Formation comprises phyllites, andalusite-cordierite mica schists, staurolite mica schists, and sillimanite mica schists, with minor quartzite and marble present.

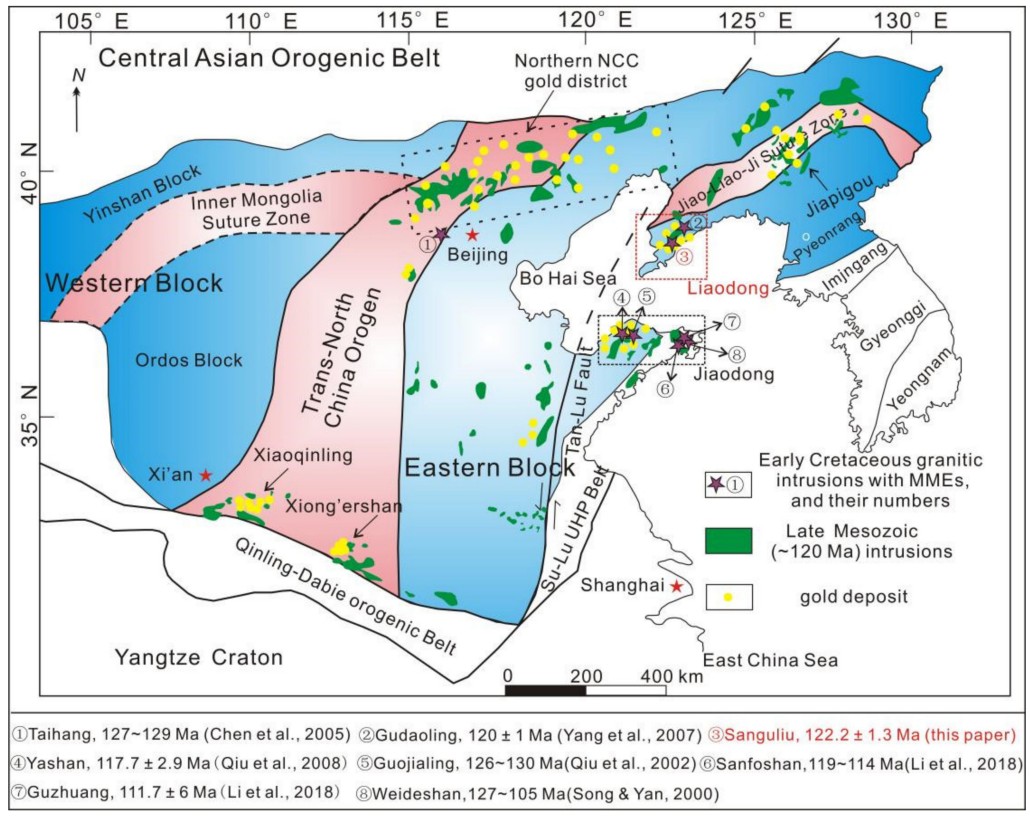

**Figure 1.** Tectonic location and division of the North China Craton (the zircon U–Pb ages from [6,8,11,20–23], modified from [24]). The locations of the mapped areas in Figure 2 have been marked by the red rectangle.

Voluminous Paleoproterozoic granitoid and mafic intrusions are also exposed in this area (Figure 2b). These mafic intrusions consist of gabbros and dolerites, most of which have been metamorphosed into greenschists and amphibolites, and the granitoid plutons are composed of deformed monzogranitic gneisses (traditionally called the Liaoji Granites), undeformed porphyritic monzogranites and granites, and alkaline syenites [26].

After the Paleoproterozoic rifting/orogenic events, this area was partially covered by Mesoproterozoic to Paleozoic metasediments (Figure 2b) [27]. During the Late Triassic, the area was affected by either the closure of the Paleo-Asian Ocean or the amalgamation of the Yangtze Craton (YC) and NCC to be featured by the local emplacement of Middle-Late Triassic granite (Figure 2b). Since the Late Mesozoic, the Liaodong Peninsula has become an important part of the circum-Pacific tectono-magmatic zone and is characterized by intensive tectonic, magmatic, and metallogenic events [28,29]. The Late Mesozoic intrusions are mainly divided into (1) Jurassic (180–153 Ma) tonalite, diorite, and gneissic two-mica monzogranite, which have experienced ductile deformation, and (2) Early Cretaceous (131–100 Ma) undeformed to slightly deformed diorite, granodiorite, monzogranite, and

syenogranite (Figure 2b) [8,30]. Accompanied by the widespread Late Mesozoic granitoids, several episodes of volcanism, terrestrial deposition, extensional structures, and gold mineralization occurred.

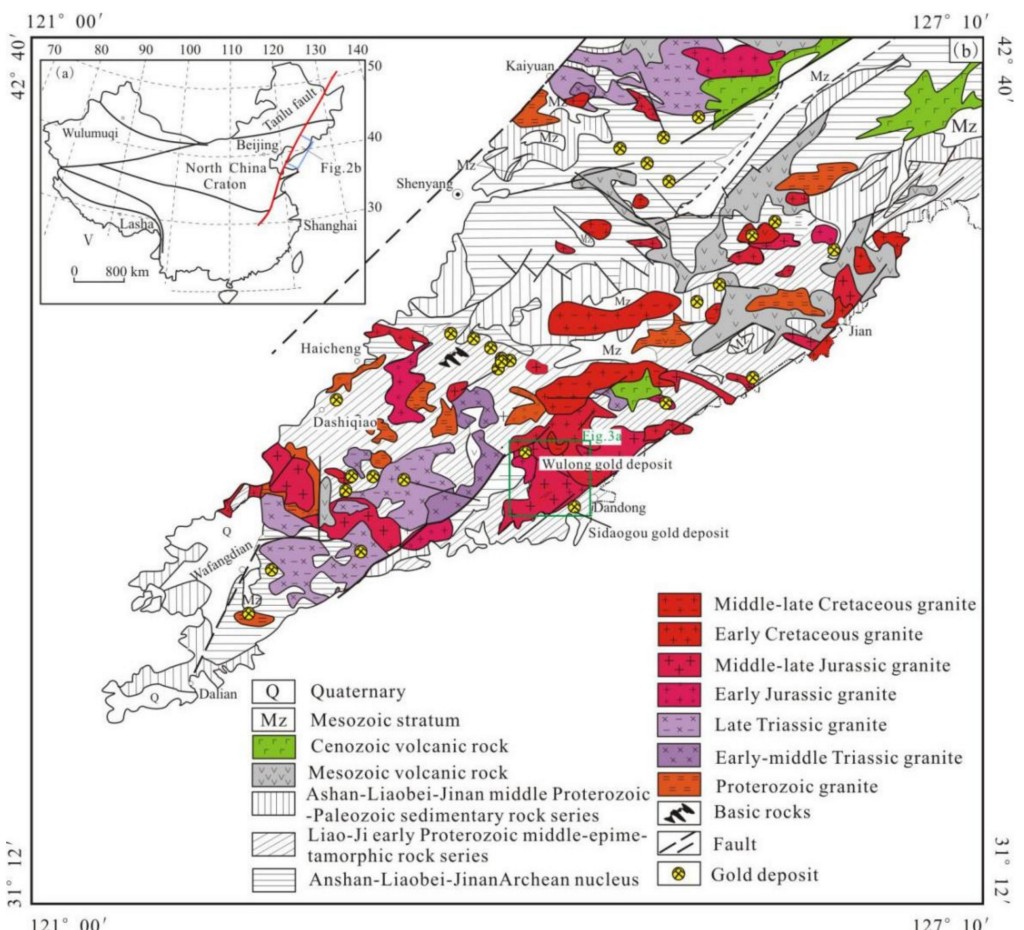

**Figure 2.** (**a**) Location map of the Liaodong Peninsula. (**b**) Sketch map showing the regional geology and distribution of gold deposits in the Liaodong Peninsula (modified from [25]).

## 3. Geology and Petrography of the Sanguliu Pluton

The Sanguliu pluton is situated in the middle of the Wulong ore field in Dandong City (Figure 3a), which contains the large tonnage of Wulong and Sidaogou gold deposits, as well as many other small gold deposits and occurrences (Figure 2a). This pluton is cut by the auriferous quartz veins of the large Wulong gold deposit (Figure 3b) and has slight sericitization and disseminated pyritization. It occurs as a stock with an outcrop of 40–50 km$^2$ intruding into the Paleoproterozoic metamorphic rocks of the Liaohe Group and Jurassic Wulong gneissic two-mica granite with a sensitive high-resolution ion micro-probe (SHRIMP) zircon U-Pb age of 158 ± 2 Ma [31] and is cut by a series of north-northeast-trending faults (Figure 3a). The Sanguliu intrusion yields obvious facies zoning, wherein its lithology changes from monzogranite in the center to granodiorite and quartz diorite at the border (Figure 3a).

With the exception of the xenoliths of the metamorphic rocks of the Liaohe Group, monzodioritic enclaves are ubiquitous in the Sanguliu intrusion and are oval or elliptical, ranging from five to forty centimeters in diameter (Figure 4a,b). In general, these enclaves have sharp contact with the host Sanguliu intrusion (Figure 4c). The petrography of the host Sanguliu granodiorite and its enclave studied herein are described as follows.

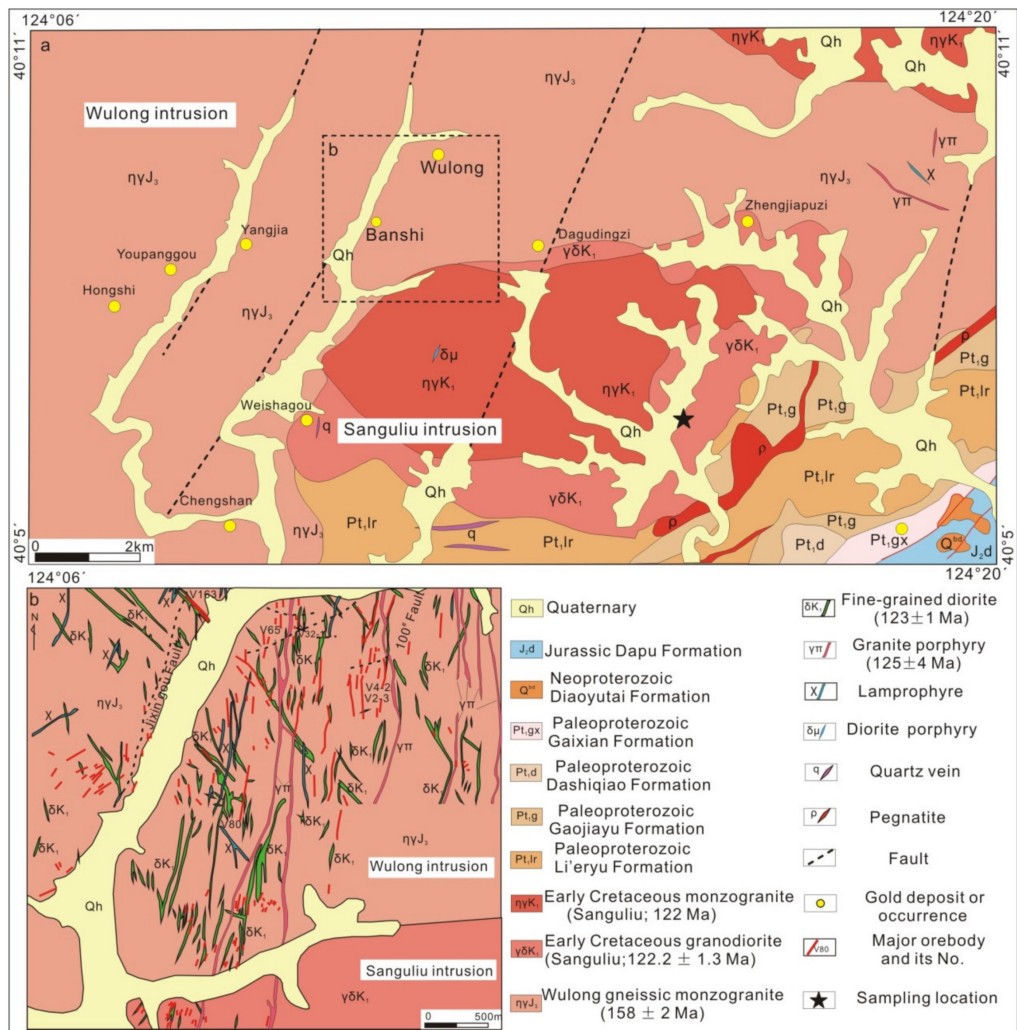

**Figure 3.** Geological maps of the Wulong gold orefield (**a**) and Wulong gold deposit (**b**) (modified from [16]. The geochronological data are from [30–32]).

The Sanguliu granodiorite has a medium–fine-grained granular texture and consists of quartz (20%–30%), plagioclase (35%–45%), potassium feldspar (15%–25%), biotite (10%–15%), and minor hornblende (approximately 5%), as well as accessory minerals of titanite and apatite (Figure 4c). Note that plagioclase commonly develops compositional zoning.

Meanwhile, the monzodiorite enclave has, in general, a hypautomorphic granular texture and comprises plagioclase (40%–45%), potassium feldspar (15%–20%), hornblende (approximately 20%), biotite (5%–10%), and minor quartz (<5%), as well as accessory minerals of apatite and zircon (Figure 4c–f). Furthermore, some plagioclase within the enclave occurs as an elliptic chadacryst that is trapped from the host granodiorite and develops resorption and zonal textures (Figure 4d,e). The apatite is always acicular with a length–width ratio of 10:1–20:1 (Figure 4f).

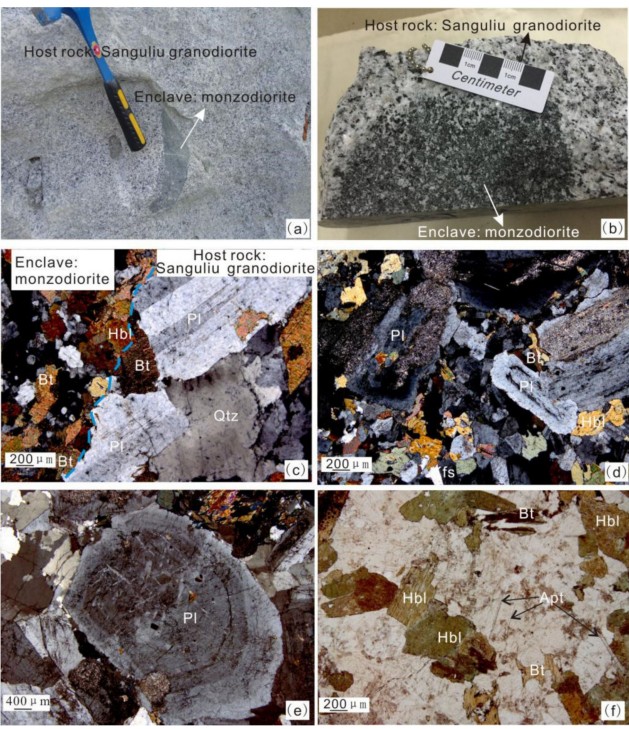

**Figure 4.** Photographs (**a**,**b**) and microphotographs (**c**–**f**) of the Sanguliu granodiorite and enclaves. (**a**,**b**) The monzodioritic enclaves occur in elliptic and marquise shape within the Sanguliu granodiorite; (**c**) The host granodiorite and enclave have sharp contact; (**d**) The plagioclase within the host granodiorite has zonal texture; (**e**,**f**) The monzodioritic enclave has plagioclase of zonal texture and acicular apatite. Apl—Apatite; Bt—Biotite; Hbl—Hornblende; Kfs—Potassium feldspar; Pl—Plagioclase; Qtz—Quartz.

## 4. Analytical Methods

### 4.1. Zircon LA–ICP–MS and SHRIMP U–Pb Isotopic Dating

Separation, target preparation, and cathodoluminescence (CL) images of zircon from Sanguliu granodiorite (sample SGL) were carried out by the Hongxin Geology and Exploration Technology Service Limited Company of Hebei Province, China, and laser ablation–inductively coupled plasma–mass spectrometry (LA–ICP–MS) zircon U–Pb dating was completed at Key Laboratory of Mineral Resources Evaluation in Northeast Asia, Ministry of Land and Resource.

Target preparation, CL images, and the SHRIMP U–Pb isotopic dating of zircon from the monzodiorite enclave (sample BTZ-1) were performed at the Beijing Ion Probe Center of the China Institute of Geological Sciences. Zircon SHRIMP U–Pb isotopic dating was completed using a SHRIMP II instrument. The primary $O^{-2}$ flow intensity and beam spot diameter were maintained at 3–5 nA and 25 μm, respectively. The standard sample M257 (U = $840 \times 10^{-6}$, [33]) and TEM (age, 417 Ma; [34]) were used to correct the zircon U content and age, respectively. The standard zircon TEM was analyzed after investigating all three unknown samples. SQUID and ISOPLOT software were used for data processing [35]. The measured $^{204}Pb$ content was used to correct for ordinary lead, and $^{207}Pb/^{206}Pb$ (more than 1000 Ma) or $^{206}Pb/^{238}U$ (less than 1000 Ma) ages were adopted for the zircon age. Details of the testing methods and experimental procedures, as well as related parameters and error corrections, were identical to those applied in Williams [36].

### 4.2. Zircon In Situ Lu–Hf Isotopes

Zircon in situ Lu–Hf isotopic analyses were conducted at the State Key Laboratory for Endogenic Metal Deposits Research, Nanjing University, using a Neptune Laser Ablation Multiple-Receiver LA–ICP–MS with a 193 nm laser sampling system. The laser spot

diameter was 60 µm, the pulse width was 15 ns, and the abrasion material carrier gas was He. Detailed analytical procedures are provided in the literature [37,38]. Measurements of the 91500 and MT zircons were conducted after every ten sample analyses. These standard samples, that is, 91500 and MT, had $^{176}$Hf/$^{177}$Hf ratios of $0.282316 \pm 30$ and $0.282507 \pm 50$, respectively. In the $\varepsilon_{Hf}(t)$ and model age calculations, the $^{176}$Lu decay constant was $1.867^{-11}$ [39], and the $^{176}$Lu/$^{177}$Hf and $^{176}$Hf/$^{177}$Hf ratios of the present-day chondritic and depleted mantle were 0.0332 and 0.282772 [40], and 0.0384 and 0.28325, respectively [41]. Additionally, the calculation of the crustal model ages determined an average crustal value of $^{176}$Lu/$^{177}$Hf = 0.015 [42].

### 4.3. Whole-Rock Geochemistry

A total of 21 fresh and unaltered samples for the Sanguliu granodiorite (16) and monzodiorite enclave (5) were selected for major, trace, and rare earth element (REE) compositions analyses, which were conducted at the Northeast China Supervision and Inspection Center of Mineral Resources, Ministry of Natural Resources of China. The major element compositions were analyzed using a Philips PW2404 type X-ray fluorescence spectrometer, which has an analytical precision of better than 5%. Meanwhile, the trace element and REE compositions were determined using a nHR–ICP–MS (Element I), which has a precision of better than 5% if the contents of trace elements were more than 10 ppm, and better than 10% if their contents were less than 10 ppm.

### 4.4. Sr–Nd–Pb Isotopes

A total of 10 fresh and unaltered samples of the Sanguliu granodiorite (5) and monzodiorite enclave (5) were selected for Rb–Sr, Sm–Nd, and Pb isotope analyses, which were conducted at the Analytical Laboratory of Beijing Research Institute of Uranium Geology using an ISOPROBE-T type thermoelectric ionization mass spectrometer (TIMS). The mass fractionations of the Rb–Sr and Sm–Nd isotopes were corrected using $^{86}$Sr/$^{88}$Sr = 0.1194 and $^{146}$Nd/$^{144}$Nd = 0.7219, respectively, with corresponding standard results of NBS987 = $0.710250 \pm 7$ and $^{143}$Nd/$^{144}$Nd = $0.512109 \pm 3$ (JMC). The experiment background values of Rb, Sr, Sm, and Nd were $2 \times 10^{-10}$ g, $2 \times 10^{-10}$ g, $<50 \times 10^{-12}$ g, and $<50 \times 10^{-12}$ g, respectively.

The Pb isotopes were identified via static acceptance and the uncorrected results of NBS981 were $^{208}$Pb/$^{206}$Pb = $2.164940 \pm 15$, $^{207}$Pb/$^{206}$Pb = $0.914338 \pm 7$, and $^{204}$Pb/$^{206}$Pb = $0.591107 \pm 2$. The experimental background Pb content was $<100 \times 10^{-12}$ g.

### 4.5. Mineral Chemical Compositions

Mineral compositions were determined at the State Key Laboratory of Geological Processes and Mineral Resources, China University of Geosciences (Wuhan), using a JEOL JXA-8100 Electron Probe Micro Analyzer (EPMA) equipped with four wavelength-dispersive spectrometers. Prior to analysis, the samples were coated with a thin conductive carbon film. Precautions suggested by Zhang and Yang [43] were employed to minimize the differences in the carbon film thickness between samples in order to obtain a uniform coating of approximately 20 nm. During the analysis, an accelerating voltage of 15 kV, beam current of 20 nA, and 10 µm spot size were used. The data were corrected online using a modified ZAF (atomic number, absorption, fluorescence) correction procedure. The peak counting times were 10s for F, Cl, Na, Mg, Al, Si, K, Ca, and Fe, and 20s for Ti and Mn. The background counting time was half of the peak counting time for the high- and low-energy background positions. The following standards were used: Fluorite (F), Halite (Cl), Sanidine (K, Al), Almandine (Fe), Diopsode (Ca, Mg), Jadeite (Na), Rhodonite (Mn), Olivine (Si), and Rutile (Ti).

## 5. Results

### 5.1. Zircon U–Pb Dating

The LA–ICP–MS and SHRIMP zircon U–Pb dating data for the granodiorite intrusion (sample SGL) and monzodioritic enclave (sample BT1) are listed in Tables 1 and 2, respectively. The analyzed zircon grains from the granodiorite and monzodiorite are euhedral–subhedral, colorless, exhibit weak or clear oscillatory growth zoning in the CL images (Figure 5), and have Th/U ratios of 0.78 to 1.47 and 0.47 to 1.79, respectively. These zircon characteristics are indicative of magmatic origins [44].

**Table 1.** Zircon LA–ICP–MS U–Pb data of the host Sanguliu granodiorite.

| Spot | Th (ppm) | U (ppm) | Th/U | $^{207}Pb/^{206}Pb$ | 1σ | $^{207}Pb/^{235}U$ | 1σ | $^{206}Pb/^{238}U$ | 1σ | $^{206}Pb/^{238}U$ Age (Ma) | 1σ |
|------|----------|---------|------|---------------------|-----|--------------------|-----|--------------------|-----|------|-----|
| SGL-1 | 314.54 | 394.93 | 0.80 | 0.04795 | 0.00217 | 0.12568 | 0.00579 | 0.01901 | 0.00048 | 121 | 3 |
| SGL-2 | 339.58 | 305.64 | 1.11 | 0.05641 | 0.00232 | 0.14493 | 0.00611 | 0.01863 | 0.00047 | 119 | 3 |
| SGL-3 | 113.06 | 97.69 | 1.16 | 0.0489 | 0.00437 | 0.12844 | 0.01139 | 0.01905 | 0.00056 | 122 | 4 |
| SGL-4 | 253.45 | 237.22 | 1.07 | 0.04802 | 0.00233 | 0.12568 | 0.00617 | 0.01898 | 0.00049 | 121 | 3 |
| SGL-5 | 268.08 | 343.25 | 0.78 | 0.04829 | 0.00208 | 0.12706 | 0.0056 | 0.01908 | 0.00048 | 122 | 3 |
| SGL-6 | 247.74 | 228.56 | 1.08 | 0.05051 | 0.00291 | 0.12866 | 0.00742 | 0.01848 | 0.0005 | 118 | 3 |
| SGL-7 | 180.27 | 207.67 | 0.87 | 0.04666 | 0.00304 | 0.12289 | 0.00798 | 0.0191 | 0.00052 | 122 | 3 |
| SGL-8 | 271.5 | 250.81 | 1.08 | 0.04914 | 0.00231 | 0.12864 | 0.00614 | 0.01899 | 0.00049 | 121 | 3 |
| SGL-9 | 130.22 | 140.73 | 0.93 | 0.05335 | 0.00326 | 0.14075 | 0.00859 | 0.01914 | 0.00052 | 122 | 3 |
| SGL-10 | 282.68 | 321.85 | 0.88 | 0.04916 | 0.00237 | 0.12945 | 0.0063 | 0.0191 | 0.00049 | 122 | 3 |
| SGL-11 | 292.1 | 319.06 | 0.92 | 0.04859 | 0.00198 | 0.12915 | 0.0054 | 0.01928 | 0.00048 | 123 | 3 |
| SGL-12 | 329.77 | 317.11 | 1.04 | 0.05033 | 0.00217 | 0.13316 | 0.00586 | 0.01919 | 0.00049 | 123 | 3 |
| SGL-13 | 108.91 | 89.63 | 1.22 | 0.07339 | 0.00433 | 0.20095 | 0.01176 | 0.01986 | 0.00056 | 127 | 4 |
| SGL-14 | 76.28 | 71.69 | 1.06 | 0.04858 | 0.0041 | 0.12803 | 0.01073 | 0.01912 | 0.00055 | 122 | 3 |
| SGL-15 | 108.26 | 98.94 | 1.09 | 0.05052 | 0.00377 | 0.13311 | 0.00986 | 0.01911 | 0.00054 | 122 | 3 |
| SGL-16 | 96.35 | 72.88 | 1.32 | 0.05013 | 0.00466 | 0.1313 | 0.01209 | 0.019 | 0.00057 | 121 | 4 |
| SGL-17 | 252.7 | 290.28 | 0.87 | 0.04837 | 0.00203 | 0.12774 | 0.00547 | 0.01915 | 0.00048 | 122 | 3 |
| SGL-18 | 83.58 | 76.59 | 1.09 | 0.0665 | 0.00423 | 0.18242 | 0.01152 | 0.0199 | 0.00056 | 127 | 4 |
| SGL-19 | 157.08 | 176.61 | 0.89 | 0.05092 | 0.0026 | 0.13154 | 0.00677 | 0.01874 | 0.00049 | 120 | 3 |
| SGL-20 | 234.14 | 219.76 | 1.07 | 0.0481 | 0.00244 | 0.12838 | 0.00656 | 0.01936 | 0.0005 | 124 | 3 |
| SGL-21 | 333.06 | 263.03 | 1.27 | 0.04828 | 0.00217 | 0.12577 | 0.00573 | 0.01889 | 0.00048 | 121 | 3 |
| SGL-22 | 67.03 | 61.7 | 1.09 | 0.04893 | 0.00433 | 0.13275 | 0.01165 | 0.01968 | 0.00058 | 126 | 4 |
| SGL-23 | 61.03 | 60.07 | 1.02 | 0.04927 | 0.00413 | 0.13373 | 0.01114 | 0.01968 | 0.00056 | 126 | 4 |
| SGL-24 | 83.76 | 57.04 | 1.47 | 0.04748 | 0.00387 | 0.12773 | 0.01034 | 0.01951 | 0.00055 | 125 | 3 |

**Table 2.** Zircon SHRIMP U-Pb data of the monzodiorite enclaves.

| Spot | $^{206}Pb_c$ (%) | U (ppm) | Th (ppm) | $^{232}Th/^{238}U$ | $^{206}Pb$ * (ppm) | $^{207}Pb$ */$^{206}Pb$ * | 1σ (%) | $^{207}Pb$ */$^{235}U$ | 1σ (%) | $^{206}Pb$ */$^{238}U$ | 1σ (%) | Errors | $^{206}Pb/^{238}U$ Age (Ma) | $^{207}Pb/^{206}Pb$ Age (Ma) |
|------|-----|-----|-----|-----|-----|-----|-----|-----|-----|-----|-----|-----|-----|-----|
| BTZ-1-1 | 0.27 | 73 | 90 | 1.28 | 1.24 | 0.055 | 20 | 0.15 | 20 | 0.01965 | 2.7 | 0.135 | 124.3 ± 3.3 | 432 ± 440 |
| BTZ-1-2 | 0.59 | 237 | 209 | 0.91 | 3.93 | 0.0483 | 8.3 | 0.128 | 8.6 | 0.01918 | 2.1 | 0.241 | 122.5 ± 2.5 | 115 ± 200 |
| BTZ-1-3 | 0.94 | 121 | 98 | 0.83 | 2.09 | 0.0505 | 14 | 0.139 | 15 | 0.01991 | 2.3 | 0.159 | 126.8 ± 2.7 | 219 ± 330 |
| BTZ-1-4 | 1.03 | 105 | 122 | 1.20 | 1.71 | 0.043 | 25 | 0.112 | 25 | 0.01864 | 2.6 | 0.104 | 119.8 ± 2.7 | −140 ± 630 |
| BTZ-1-5 | 1.55 | 98 | 94 | 0.99 | 1.7 | 0.0445 | 14 | 0.122 | 14 | 0.01989 | 2.4 | 0.166 | 127.6 ± 2.9 | −80 ± 350 |
| BTZ-1-6 | 0.56 | 371 | 288 | 0.80 | 6.33 | 0.0449 | 7.8 | 0.1223 | 8 | 0.01975 | 2 | 0.252 | 126.6 ± 2.5 | −60 ± 190 |
| BTZ-1-7 | 1.41 | 286 | 358 | 1.29 | 6.78 | 0.2535 | 3.3 | 0.951 | 3.9 | 0.02721 | 2 | 0.523 | 129.1 ± 3.1 | 3207 ± 52 |
| BTZ-1-8 | 1.66 | 96 | 100 | 1.07 | 1.65 | 0.0455 | 16 | 0.123 | 16 | 0.0196 | 2.9 | 0.176 | 125.6 ± 3.5 | −29 ± 390 |
| BTZ-1-9 | 0.41 | 394 | 269 | 0.71 | 6.89 | 0.0475 | 5.6 | 0.1328 | 5.9 | 0.02027 | 2 | 0.338 | 129.6 ± 2.5 | 76 ± 130 |
| BTZ-1-10 | 0.65 | 160 | 277 | 1.79 | 2.74 | 0.0462 | 14 | 0.126 | 14 | 0.0198 | 2.2 | 0.156 | 126.7 ± 2.7 | 9 ± 340 |
| BTZ-1-11 | 1.51 | 108 | 172 | 1.65 | 1.79 | 0.0391 | 18 | 0.103 | 19 | 0.0191 | 2.5 | 0.132 | 123.4 ± 2.8 | −413 ± 480 |
| BTZ-1-12 | 0.10 | 179 | 89 | 0.51 | 45.2 | 0.1086 | 1.3 | 4.4 | 2.4 | 0.294 | 2 | 0.838 | 1649 ± 32 | 1776 ± 24 |
| BTZ-1-13 | 0.86 | 158 | 139 | 0.91 | 2.75 | 0.0447 | 15 | 0.124 | 16 | 0.02005 | 2.3 | 0.145 | 128.6 ± 2.7 | −72 ± 380 |
| BTZ-1-14 | 0.95 | 266 | 204 | 0.79 | 4.49 | 0.0427 | 5.4 | 0.1146 | 5.7 | 0.01945 | 2 | 0.353 | 125.1 ± 2.5 | −183 ± 130 |

Note: Errors = 1 σ; Pb and Pb indicate common and radiogenic portions, respectively. Common Pb corrected using measured Pb.

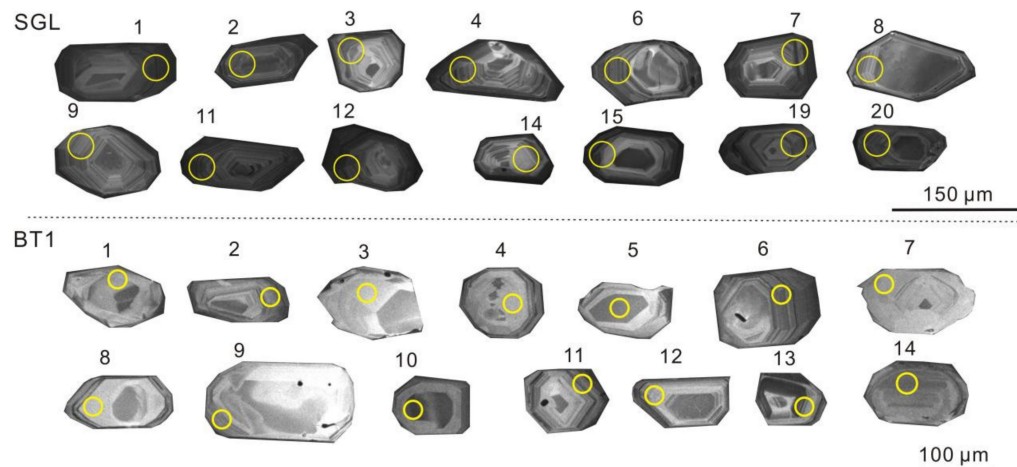

**Figure 5.** Cathodoluminescence (CL) images of zircons from the host Sanguliu granodiorite and monzodioritic enclaves.

Twenty-four granodiorite zircon analyses formed a relatively tight concordia cluster (Figure 6a) and their $^{206}$Pb/$^{238}$U ages ranged from 118 ± 3 to 127 ± 4 Ma, with a weighted mean age of 122.2 ± 1.3 Ma (MSWD = 0.45, N = 24) (Figure 6b), which may represent the final emplacement age of the host granodiorite.

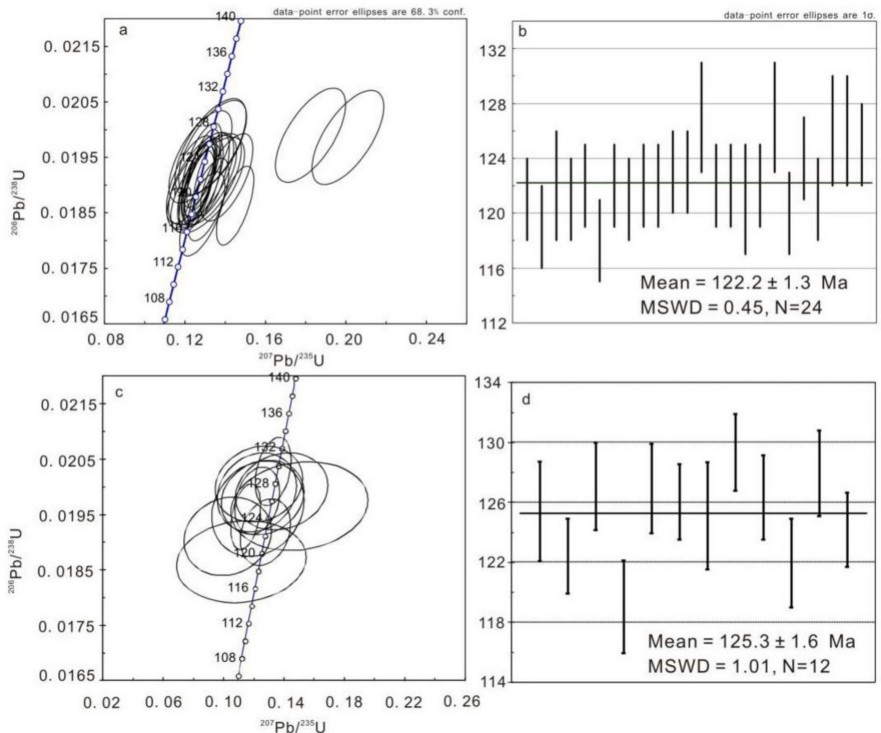

**Figure 6.** LA–ICP–MS and SHRIMP zircon U–Pb concordia and weighted mean age diagrams of the Sanguliu granodiorite (**a**,**b**) and monzodioritic enclave (**c**,**d**).

Similarly, thirteen zircons' analyses from the monzodioritic enclaves resulted in a tight cluster on a concordia (Figure 6c) and their $^{206}$Pb/$^{238}$U ages range from 119.1 ± 3.1 to 129.4 ± 2.6 Ma, with a weighted mean age of 125.3 ± 1.6 Ma (MSWD = 1.01, N = 12) (Figure 6d), which could be considered to be the crystallization age of the monzodioritic enclave. Furthermore, one zircon spot yielded an older $^{207}$Pb/$^{206}$Pb age of 1776 ± 24 Ma, which may have been inherited from the Paleoproterozoic Liaohe Group.

### 5.2. Zircon In Situ Lu–Hf Isotopes

Lu–Hf isotopic analyses were conducted on the 13 zircon grains with concordant SHRIMP U–Pb ages, and the results are listed in Table 3. The $^{176}Lu/^{177}Hf$ ratios for 13 spots ranged from 0.000362 to 0.00125, with $\varepsilon_{Hf}(t)$ values ranging from −18.6 to −22.7, and $T_{DM2}$ model ages of 2404 to 2613 Ma (Figure 7).

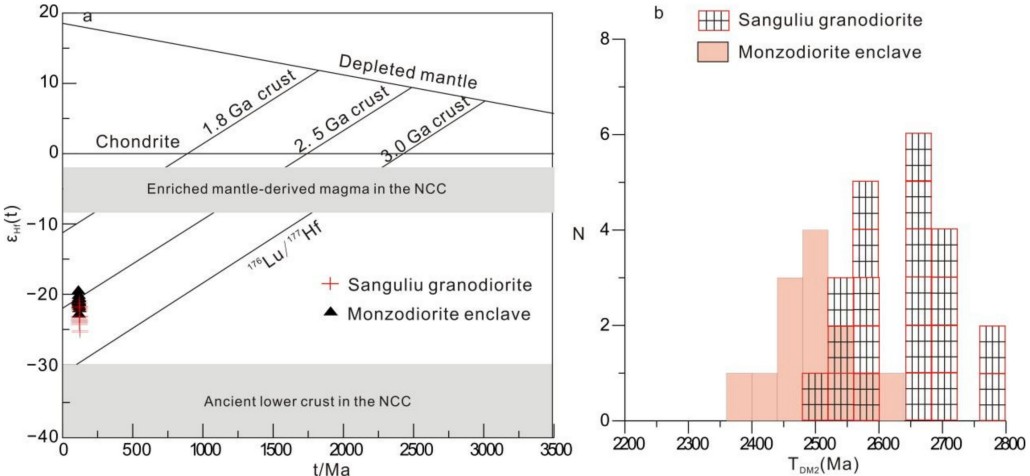

**Figure 7.** Evolution $\varepsilon_{Hf}(t)$ from in situ analysis of zircon Hf versus age (Ma, (**a**)) and histogram of $T_{DM2}$ model ages (Ma, (**b**)) of the Sanguliu granodiorite and monzodioritic enclaves. The data of enriched mantle-derived magma and ancient lower crust in the NCC, the Sanguliu granodiorite, as well as enclaves are from [45–47], respectively.

### 5.3. Major and Trace Elements

Major and trace elemental data for the Sanguliu granodiorite and monzodioritic enclave are listed in Table 4. The Sanguliu granodiorite has medium $SiO_2$ (67.54 to 68.45 wt.%), medium $Al_2O_3$ (14.82 to 15.39 wt.%), $K_2O$ (4.20 to 4.40 wt.%), and low MgO (1.51 to 1.72 wt.%) and $Na_2O$ (3.22 to 3.46 wt.%) contents, with total alkalis and $Na_2O/K_2O$ ratios ranging from 7.50 to 7.86 wt.%, and from 0.75 to 0.82, respectively. It also has medium values of aluminum saturation index (A/CNK = 0.92–1.02) and $Mg^{\#}$ values ranging from 43.73 to 47.40. Conversely, the monzodioritic enclave has low $SiO_2$ (54.17 to 55.78 wt.%) and $Na_2O$ (2.15 to 4.67 wt.%), variable $Al_2O_3$ (12.92 to 17.07 wt.%) and $K_2O$ (1.93 to 5.95 wt.%), and high MgO (4.18 to 6.17 wt.%) and CaO (5.45 to 6.53 wt.%) contents, with total alkalis contents of 6.56 to 8.32 wt.%. The monzodioritic enclave also exhibits low values of aluminum saturation index (A/CNK = 0.62–0.84), and $Mg^{\#}$ values ranging from 49.76 to 56.62. In the total alkali–silica diagram, the Sanguliu granodiorite and monzodiorite enclave samples fell into the sub-alkalic granodiorite and alkalic monzodiorite-monzonite fields, respectively (Figure 8a), while the Sanguliu granodiorite samples plotted in the high-K calc-alkaline field in the $SiO_2$–$K_2O$ diagram (Figure 8b), and all the evaluated samples plotted into the metaluminous field of the A/NK–A/CNK diagram (Figure 8c). These results indicate that the Sanguliu granodiorite is high-K calc-alkaline and metaluminous, whereas the monzodiorite is alkalic and metaluminous.

**Table 3.** Zircon in situ Lu–Hf isotopic data for the host Sanguliu granodiorite and monzodiorite enclaves.

| Sample | SHRIMP Age (Ma) | $^{176}$Yb/$^{177}$Hf | 2σ | $^{176}$Lu/$^{177}$Hf | 2σ | $^{176}$Hf/$^{177}$Hf | 2σ | $^{176}$Hf/$^{177}$Hf$_i$ | e$_{Hf}$(0) | e$_{Hf}$(t) | 2σ | T$_{DM}$ (Ma) | T$_{DM}$$^C$ (Ma) | f$_{Lu/Hf}$ |
|---|---|---|---|---|---|---|---|---|---|---|---|---|---|---|
| BTZ-1-1 | 124.3 | 0.033152 | 0.000545 | 0.001250 | 0.000021 | 0.282056 | 0.000019 | 0.282053 | −25.3 | −22.7 | 0.7 | 1697 | 2614 | −0.96 |
| BTZ-1-2 | 122.5 | 0.012728 | 0.000112 | 0.000462 | 0.000004 | 0.282094 | 0.000019 | 0.282093 | −24.0 | −21.3 | 0.7 | 1610 | 2527 | −0.99 |
| BTZ-1-3 | 126.8 | 0.010377 | 0.000069 | 0.000362 | 0.000002 | 0.282114 | 0.000018 | 0.282113 | −23.3 | −20.5 | 0.6 | 1578 | 2479 | −0.99 |
| BTZ-1-4 | 119.8 | 0.012539 | 0.000112 | 0.000458 | 0.000004 | 0.282110 | 0.000020 | 0.282109 | −23.4 | −20.8 | 0.7 | 1587 | 2493 | −0.99 |
| BTZ-1-5 | 127.6 | 0.012572 | 0.000121 | 0.000452 | 0.000005 | 0.282104 | 0.000018 | 0.282103 | −23.6 | −20.9 | 0.6 | 1595 | 2501 | −0.99 |
| BTZ-1-6 | 126.6 | 0.017442 | 0.000028 | 0.000655 | 0.000001 | 0.282102 | 0.000016 | 0.282101 | −23.7 | −21.0 | 0.6 | 1606 | 2507 | −0.98 |
| BTZ-1-7 | 129.1 | 0.015908 | 0.000368 | 0.000580 | 0.000013 | 0.282140 | 0.000019 | 0.282139 | −22.3 | −19.6 | 0.7 | 1551 | 2420 | −0.98 |
| BTZ-1-8 | 125.6 | 0.010517 | 0.000187 | 0.000381 | 0.000006 | 0.282099 | 0.000018 | 0.282099 | −23.8 | −21.1 | 0.6 | 1599 | 2512 | −0.99 |
| BTZ-1-9 | 129.6 | 0.013294 | 0.000063 | 0.000485 | 0.000002 | 0.282074 | 0.000015 | 0.282072 | −24.7 | −21.9 | 0.5 | 1638 | 2568 | −0.99 |
| BTZ-1-10 | 126.7 | 0.019775 | 0.000241 | 0.000662 | 0.000007 | 0.282119 | 0.000019 | 0.282117 | −23.1 | −20.4 | 0.7 | 1583 | 2470 | −0.98 |
| BTZ-1-11 | 123.4 | 0.017998 | 0.000345 | 0.000603 | 0.000011 | 0.282150 | 0.000017 | 0.282148 | −22.0 | −19.4 | 0.6 | 1539 | 2403 | −0.98 |
| BTZ-1-13 | 128.6 | 0.014534 | 0.000412 | 0.000523 | 0.000016 | 0.282084 | 0.000019 | 0.282082 | −24.3 | −21.6 | 0.7 | 1626 | 2546 | −0.98 |
| BTZ-1-14 | 125.1 | 0.025354 | 0.000784 | 0.000968 | 0.000031 | 0.282131 | 0.000020 | 0.282128 | −22.7 | −20.0 | 0.7 | 1580 | 2446 | −0.97 |
| TW005-1.1 | 125.1 | 0.006358 | 0.000046 | 0.000274 | 0.000002 | 0.282065 | 0.000020 | 0.282042 | −25.8 | −23.1 | 0.7 | 1672 | 2642 | −0.99 |
| TW005-10.1 | 121.8 | 0.006722 | 0.000068 | 0.000280 | 0.000003 | 0.282105 | 0.000018 | 0.282082 | −24.4 | −21.7 | 0.6 | 1618 | 2555 | −0.99 |
| TW005-11.1 | 120.6 | 0.009948 | 0.000047 | 0.000420 | 0.000003 | 0.282053 | 0.000019 | 0.282030 | −26.2 | −23.6 | 0.7 | 1695 | 2672 | −0.99 |
| TW005-12.1 | 121.2 | 0.011933 | 0.000055 | 0.000500 | 0.000002 | 0.282064 | 0.000019 | 0.282041 | −25.8 | −23.2 | 0.7 | 1683 | 2647 | −0.98 |
| TW005-15.1 | 121.9 | 0.010452 | 0.000095 | 0.000439 | 0.000004 | 0.282058 | 0.000019 | 0.282035 | −26.1 | −23.4 | 0.7 | 1689 | 2660 | −0.99 |
| TW005-3.1 | 122.2 | 0.008121 | 0.000020 | 0.000340 | 0.000001 | 0.282096 | 0.000020 | 0.282073 | −24.7 | −22.1 | 0.7 | 1633 | 2576 | −0.99 |
| TW005-4.1 | 123.9 | 0.010637 | 0.000063 | 0.000442 | 0.000003 | 0.282107 | 0.000023 | 0.282084 | −24.3 | −21.6 | 0.8 | 1622 | 2550 | −0.99 |
| TW005-5.1 | 120.4 | 0.012798 | 0.000243 | 0.000519 | 0.000009 | 0.282121 | 0.000021 | 0.282098 | −23.8 | −21.2 | 0.7 | 1606 | 2521 | −0.98 |
| TW005-6.1 | 126.6 | 0.009011 | 0.000039 | 0.000371 | 0.000002 | 0.282097 | 0.000021 | 0.282074 | −24.7 | −21.9 | 0.7 | 1633 | 2571 | −0.99 |
| TW005-7.1 | 123.3 | 0.010010 | 0.000221 | 0.000411 | 0.000008 | 0.282130 | 0.000023 | 0.282107 | −23.5 | −20.9 | 0.8 | 1590 | 2501 | −0.99 |
| TW005-9.1 | 125.8 | 0.012347 | 0.000106 | 0.000492 | 0.000004 | 0.282101 | 0.000028 | 0.282078 | −24.5 | −21.8 | 1.0 | 1632 | 2563 | −0.99 |
| TW006-1.1 | 125.5 | 0.012540 | 0.000146 | 0.000524 | 0.000006 | 0.282037 | 0.000020 | 0.282014 | −26.8 | −24.1 | 0.7 | 1721 | 2704 | −0.98 |
| TW006-11.1 | 121.2 | 0.011237 | 0.000370 | 0.000467 | 0.000016 | 0.282035 | 0.000022 | 0.282012 | −26.9 | −24.3 | 0.8 | 1722 | 2713 | −0.99 |
| TW006-13.1 | 121.8 | 0.010705 | 0.000104 | 0.000441 | 0.000005 | 0.282051 | 0.000021 | 0.282028 | −26.3 | −23.7 | 0.7 | 1698 | 2675 | −0.99 |
| TW006-15.1 | 126.6 | 0.010719 | 0.000234 | 0.000445 | 0.000010 | 0.282044 | 0.000022 | 0.282021 | −26.6 | −23.8 | 0.8 | 1709 | 2689 | −0.99 |
| TW006-2.1 | 123.7 | 0.012080 | 0.000040 | 0.000510 | 0.000002 | 0.282063 | 0.000019 | 0.282041 | −25.9 | −23.2 | 0.7 | 1685 | 2648 | −0.98 |
| TW006-3.1 | 123.3 | 0.015431 | 0.000257 | 0.000642 | 0.000012 | 0.282089 | 0.000042 | 0.282066 | −25.0 | −22.3 | 1.5 | 1655 | 2591 | −0.98 |
| TW006-5.1 | 122.8 | 0.009828 | 0.000094 | 0.000401 | 0.000004 | 0.282088 | 0.000020 | 0.282065 | −25.0 | −22.3 | 0.7 | 1646 | 2593 | −0.99 |
| TW006-6.1 | 126.2 | 0.015092 | 0.000318 | 0.000627 | 0.000014 | 0.282008 | 0.000024 | 0.281985 | −27.8 | −25.1 | 0.9 | 1766 | 2769 | −0.98 |
| TW006-8.1 | 120.6 | 0.014681 | 0.000078 | 0.000609 | 0.000004 | 0.282045 | 0.000021 | 0.282022 | −26.5 | −23.9 | 0.7 | 1715 | 2691 | −0.98 |
| TW006-9.1 | 125.3 | 0.010371 | 0.000338 | 0.000426 | 0.000014 | 0.282003 | 0.000020 | 0.281980 | −28.0 | −25.3 | 0.7 | 1764 | 2781 | −0.99 |

Note: The Hf isotopic data of the host Sanguliu granodiorite (TW005 and TW006) are from [45].

**Table 4.** Major (wt.%) and trace element (ppm) geochemical data for the host Sanguliu granodiorite and monzodiorite enclaves.

| Sample / Element | SGL-1 | SGL-2 | SGL-3 | SGL-4 | SGL-5 | BT-1 | BT-2 | BT-3 | BT-4 | BT-5 | BT-6 | BT-7 | BT-8 | BT-9 | BT-10 | BT-11 | BT-12 | BT-13 | BT-14 | BT-15 | BT-16 |
|---|---|---|---|---|---|---|---|---|---|---|---|---|---|---|---|---|---|---|---|---|---|
| Na$_2$O | 3.45 | 3.46 | 3.46 | 3.22 | 3.43 | 3.87 | 4.24 | 4.37 | 4.21 | 4.41 | 2.43 | 4.46 | 4.61 | 4.67 | 4.63 | 4.39 | 4.37 | 4.60 | 3.02 | 2.39 | 2.15 |
| MgO | 1.59 | 1.64 | 1.51 | 1.53 | 1.72 | 4.82 | 4.91 | 4.60 | 4.41 | 4.62 | 6.17 | 4.87 | 4.40 | 4.82 | 5.54 | 4.31 | 4.18 | 4.20 | 6.03 | 5.95 | 5.91 |
| Al$_2$O$_3$ | 14.82 | 14.82 | 14.84 | 15.39 | 15.09 | 15.90 | 16.54 | 16.91 | 16.77 | 16.91 | 14.10 | 16.70 | 17.07 | 16.70 | 16.50 | 16.86 | 16.89 | 17.03 | 14.45 | 12.92 | 13.50 |
| SiO$_2$ | 68.02 | 67.87 | 68.45 | 67.87 | 67.54 | 55.10 | 54.17 | 55.03 | 55.73 | 54.65 | 52.59 | 54.92 | 55.24 | 54.69 | 53.68 | 55.65 | 55.78 | 55.72 | 54.61 | 55.74 | 54.91 |
| P$_2$O$_5$ | 0.11 | 0.11 | 0.11 | 0.10 | 0.11 | 0.20 | 0.21 | 0.19 | 0.18 | 0.18 | 0.19 | 0.21 | 0.22 | 0.22 | 0.21 | 0.17 | 0.17 | 0.18 | 0.12 | 0.15 | 0.13 |
| K$_2$O | 4.27 | 4.20 | 4.40 | 4.28 | 4.40 | 3.61 | 2.66 | 2.97 | 3.41 | 2.80 | 5.89 | 2.44 | 2.61 | 2.05 | 1.93 | 3.03 | 3.25 | 2.56 | 4.15 | 5.11 | 5.95 |
| CaO | 3.20 | 3.20 | 2.95 | 2.81 | 3.21 | 5.64 | 6.10 | 5.98 | 5.80 | 6.03 | 5.70 | 5.80 | 5.45 | 6.27 | 6.48 | 5.72 | 5.56 | 5.76 | 6.28 | 6.35 | 6.53 |
| TiO$_2$ | 0.48 | 0.52 | 0.48 | 0.47 | 0.49 | 0.89 | 0.94 | 0.88 | 0.88 | 0.90 | 0.90 | 0.94 | 0.98 | 0.92 | 0.85 | 0.82 | 0.82 | 0.83 | 0.65 | 0.90 | 0.81 |
| MnO | 0.06 | 0.06 | 0.06 | 0.06 | 0.06 | 0.15 | 0.15 | 0.14 | 0.14 | 0.14 | 0.18 | 0.14 | 0.13 | 0.15 | 0.15 | 0.14 | 0.13 | 0.14 | 0.18 | 0.18 | 0.18 |
| Fe$_2$O$_3$ | 1.13 | 0.55 | 0.41 | 0.41 | 0.48 | 1.18 | 1.39 | 1.32 | 1.26 | 1.28 | 1.88 | 1.37 | 1.49 | 1.36 | 1.33 | 1.53 | 1.47 | 1.31 | 1.65 | 1.44 | 1.38 |
| FeO | 2.52 | 2.97 | 2.88 | 3.14 | 2.97 | 7.28 | 7.19 | 6.47 | 6.20 | 6.47 | 8.27 | 6.96 | 6.56 | 6.87 | 7.03 | 6.38 | 6.02 | 6.29 | 7.23 | 7.28 | 6.83 |
| LOI | 0.38 | 0.44 | 0.63 | 0.48 | 0.52 | 1.11 | 1.10 | 1.12 | 1.13 | 1.19 | 1.28 | 1.07 | 1.05 | 1.24 | 1.31 | 0.87 | 1.01 | 1.00 | 1.37 | 1.37 | 1.35 |
| SUM | 100.02 | 99.85 | 100.18 | 99.76 | 100.03 | 99.73 | 99.59 | 99.97 | 100.12 | 99.57 | 99.57 | 99.89 | 99.80 | 99.95 | 99.64 | 99.85 | 99.64 | 99.62 | 99.74 | 99.77 | 99.62 |
| Na$_2$O + K$_2$O | 7.72 | 7.66 | 7.86 | 7.50 | 7.83 | 7.48 | 6.90 | 7.34 | 7.62 | 7.21 | 8.32 | 6.90 | 7.22 | 6.72 | 6.56 | 7.42 | 7.62 | 7.16 | 7.17 | 7.50 | 8.10 |
| Na$_2$O/K$_2$O | 0.81 | 0.82 | 0.79 | 0.75 | 0.78 | 1.07 | 1.59 | 1.47 | 1.23 | 1.58 | 0.41 | 1.83 | 1.77 | 2.28 | 2.40 | 1.45 | 1.34 | 1.80 | 0.73 | 0.47 | 0.36 |
| A/NK | 1.44 | 1.45 | 1.42 | 1.55 | 1.45 | 1.55 | 1.68 | 1.63 | 1.58 | 1.64 | 1.36 | 1.67 | 1.64 | 1.69 | 1.70 | 1.61 | 1.58 | 1.65 | 1.53 | 1.37 | 1.35 |
| A/CNK | 0.92 | 0.92 | 0.94 | 1.02 | 0.93 | 0.78 | 0.79 | 0.80 | 0.79 | 0.80 | 0.68 | 0.81 | 0.84 | 0.78 | 0.77 | 0.81 | 0.81 | 0.82 | 0.69 | 0.62 | 0.62 |
| Mg$^{\#}$ | 44.48 | 45.76 | 45.31 | 43.73 | 47.40 | 50.74 | 50.90 | 51.71 | 51.73 | 51.93 | 52.47 | 51.44 | 49.81 | 51.49 | 54.55 | 49.76 | 50.36 | 50.06 | 55.22 | 55.29 | 56.62 |
| Ba | 989.91 | 940.11 | 952.40 | 1000.00 | 1000.00 | 721.28 | 539.31 | 562.96 | 618.55 | 519.33 | 1600.00 | 567.97 | 565.53 | 531.14 | 483.39 | 566.21 | 634.77 | 481.85 | 987.97 | 1300.00 | 1800.00 |
| Cr | 59.04 | 59.20 | 54.60 | 43.02 | 45.92 | 130.54 | 112.59 | 73.85 | 75.83 | 79.05 | 230.02 | 97.34 | 74.06 | 101.45 | 104.43 | 53.61 | 51.77 | 45.91 | 247.62 | 290.04 | 253.74 |
| Ga | 17.27 | 17.87 | 16.87 | 17.96 | 17.26 | 23.34 | 24.99 | 23.60 | 22.41 | 24.31 | 19.43 | 24.86 | 26.14 | 25.89 | 25.81 | 24.75 | 25.31 | 25.45 | 18.96 | 17.84 | 16.06 |
| Nb | 10.11 | 11.20 | 10.84 | 10.46 | 10.51 | 18.93 | 16.26 | 13.52 | 14.23 | 13.69 | 11.36 | 18.10 | 19.09 | 18.12 | 16.65 | 20.80 | 18.41 | 19.04 | 12.97 | 18.05 | 12.96 |
| Pb | 24.08 | 25.83 | 23.93 | 25.30 | 26.34 | 24.12 | 19.48 | 24.79 | 25.06 | 23.75 | 32.35 | 19.57 | 18.52 | 21.60 | 21.31 | 24.63 | 26.20 | 21.90 | 29.05 | 30.00 | 34.32 |
| Rb | 144.85 | 156.41 | 168.43 | 175.55 | 154.60 | 187.28 | 168.28 | 162.14 | 166.57 | 161.04 | 266.68 | 165.08 | 174.25 | 133.00 | 125.83 | 154.96 | 158.27 | 140.98 | 186.87 | 181.20 | 215.11 |
| Sr | 383.88 | 385.99 | 371.63 | 378.42 | 397.14 | 276.09 | 291.76 | 309.92 | 317.66 | 315.11 | 261.95 | 314.28 | 331.28 | 309.25 | 308.86 | 297.53 | 301.49 | 287.52 | 271.62 | 236.82 | 305.79 |
| Zr | 245.26 | 257.54 | 236.68 | 267.02 | 241.48 | 246.84 | 301.37 | 308.43 | 296.56 | 300.95 | 179.14 | 206.22 | 240.36 | 219.86 | 321.37 | 286.55 | 273.99 | 230.20 | 143.61 | 82.61 | 149.12 |
| Li | 15.94 | 18.83 | 17.97 | 17.98 | 16.98 | 26.02 | 29.53 | 27.79 | 26.75 | 28.45 | 32.66 | 30.19 | 30.98 | 26.48 | 25.67 | 28.48 | 29.26 | 29.57 | 25.34 | 17.07 | 16.03 |
| Be | 2.10 | 2.19 | 2.20 | 2.06 | 2.03 | 3.51 | 3.64 | 3.29 | 3.11 | 3.33 | 1.94 | 3.96 | 3.71 | 3.77 | 3.99 | 3.63 | 3.79 | 3.98 | 2.48 | 2.13 | 1.55 |
| Sc | 9.45 | 9.37 | 7.77 | 6.93 | 9.05 | 30.47 | 30.52 | 27.80 | 25.41 | 28.30 | 21.74 | 25.34 | 20.67 | 28.27 | 30.69 | 26.31 | 26.06 | 26.89 | 26.43 | 33.84 | 29.86 |
| Co | 7.96 | 9.16 | 8.25 | 7.67 | 8.95 | 19.60 | 20.02 | 18.05 | 17.34 | 18.20 | 26.28 | 19.47 | 17.68 | 18.02 | 18.66 | 17.39 | 17.79 | 18.02 | 21.93 | 21.97 | 23.31 |
| Cs | 1.59 | 2.80 | 1.89 | 1.87 | 2.01 | 3.21 | 3.12 | 2.60 | 2.63 | 2.63 | 4.44 | 3.65 | 3.51 | 2.34 | 2.27 | 2.16 | 2.16 | 2.14 | 1.99 | 2.62 | 2.47 |
| Ni | 12.08 | 13.18 | 11.79 | 11.07 | 13.42 | 27.25 | 24.49 | 20.58 | 19.65 | 19.87 | 88.18 | 28.78 | 25.60 | 22.90 | 24.55 | 21.99 | 22.24 | 22.55 | 52.08 | 58.28 | 68.65 |
| U | 3.29 | 2.83 | 2.75 | 3.14 | 2.35 | 2.03 | 2.84 | 2.76 | 2.93 | 3.06 | 12.84 | 1.82 | 1.82 | 1.21 | 1.45 | 1.33 | 1.61 | 1.55 | 1.21 | 2.64 | 2.19 |
| Hf | 7.79 | 7.71 | 7.09 | 7.55 | 7.51 | 8.66 | 7.90 | 9.35 | 8.79 | 9.05 | 5.46 | 6.43 | 7.57 | 7.44 | 9.09 | 8.07 | 8.56 | 8.98 | 4.86 | 3.67 | 4.38 |
| Ta | 1.14 | 1.09 | 0.94 | 0.96 | 0.94 | 0.57 | 0.59 | 0.46 | 0.48 | 0.53 | 0.51 | 0.66 | 0.75 | 0.61 | 0.50 | 0.85 | 0.84 | 0.84 | 0.37 | 0.67 | 0.46 |
| Th | 29.84 | 25.98 | 25.63 | 26.10 | 24.54 | 6.55 | 8.81 | 7.70 | 9.94 | 8.02 | 14.76 | 4.90 | 6.14 | 3.46 | 4.08 | 4.08 | 4.23 | 3.85 | 2.59 | 3.77 | 2.75 |
| La | 70.71 | 59.78 | 62.02 | 63.90 | 60.76 | 18.71 | 25.97 | 27.23 | 30.92 | 27.71 | 61.80 | 22.02 | 24.69 | 17.57 | 19.30 | 19.10 | 19.02 | 17.74 | 17.43 | 13.79 | 14.45 |
| Ce | 125.43 | 107.05 | 107.71 | 111.22 | 106.73 | 40.78 | 52.01 | 51.64 | 58.75 | 51.42 | 111.33 | 44.01 | 48.29 | 37.16 | 40.04 | 39.49 | 39.48 | 37.17 | 37.51 | 33.98 | 33.56 |
| Pr | 14.10 | 12.30 | 11.85 | 12.47 | 11.96 | 5.36 | 6.38 | 6.16 | 6.96 | 6.23 | 12.90 | 5.63 | 6.18 | 5.08 | 5.35 | 5.20 | 5.09 | 4.86 | 5.33 | 5.28 | 5.09 |
| Nd | 47.10 | 41.43 | 40.00 | 41.45 | 39.89 | 20.95 | 23.31 | 22.51 | 24.97 | 23.22 | 44.96 | 21.86 | 23.07 | 20.56 | 21.70 | 19.55 | 19.16 | 18.18 | 21.95 | 23.53 | 23.05 |
| Sm | 6.88 | 6.12 | 5.70 | 5.94 | 5.67 | 3.89 | 4.13 | 4.05 | 4.32 | 4.16 | 6.71 | 4.02 | 4.30 | 4.21 | 4.35 | 3.65 | 3.46 | 3.35 | 4.57 | 5.69 | 5.57 |
| Eu | 1.26 | 1.22 | 1.24 | 1.27 | 1.25 | 0.95 | 0.92 | 0.91 | 0.98 | 0.92 | 1.29 | 1.02 | 0.96 | 0.86 | 0.87 | 0.93 | 0.91 | 0.88 | 1.05 | 1.22 | 1.44 |
| Gd | 5.98 | 5.38 | 5.10 | 5.14 | 4.95 | 3.51 | 3.59 | 3.55 | 3.76 | 3.67 | 5.87 | 3.51 | 3.66 | 3.60 | 3.83 | 3.16 | 3.07 | 3.02 | 3.81 | 4.80 | 4.54 |
| Tb | 0.79 | 0.72 | 0.68 | 0.69 | 0.65 | 0.58 | 0.56 | 0.54 | 0.56 | 0.57 | 0.78 | 0.57 | 0.56 | 0.60 | 0.62 | 0.51 | 0.49 | 0.49 | 0.65 | 0.84 | 0.80 |
| Dy | 3.97 | 3.69 | 3.40 | 3.50 | 3.25 | 3.14 | 2.91 | 2.88 | 2.87 | 3.06 | 3.90 | 3.00 | 3.01 | 3.26 | 3.37 | 2.75 | 2.69 | 2.68 | 3.60 | 4.76 | 4.38 |

**Table 4.** *Cont.*

| Sample / Element | SGL-1 | SGL-2 | SGL-3 | SGL-4 | SGL-5 | BT-1 | BT-2 | BT-3 | BT-4 | BT-5 | BT-6 | BT-7 | BT-8 | BT-9 | BT-10 | BT-11 | BT-12 | BT-13 | BT-14 | BT-15 | BT-16 |
|---|---|---|---|---|---|---|---|---|---|---|---|---|---|---|---|---|---|---|---|---|---|
| Ho | 0.77 | 0.72 | 0.67 | 0.68 | 0.63 | 0.65 | 0.61 | 0.58 | 0.58 | 0.61 | 0.75 | 0.63 | 0.63 | 0.66 | 0.67 | 0.59 | 0.58 | 0.56 | 0.72 | 0.92 | 0.87 |
| Er | 2.17 | 1.97 | 1.86 | 1.90 | 1.74 | 1.90 | 1.79 | 1.62 | 1.69 | 1.70 | 2.02 | 1.75 | 1.72 | 1.80 | 1.88 | 1.72 | 1.70 | 1.64 | 2.00 | 2.40 | 2.22 |
| Tm | 0.34 | 0.32 | 0.30 | 0.31 | 0.28 | 0.33 | 0.31 | 0.28 | 0.28 | 0.28 | 0.32 | 0.30 | 0.30 | 0.31 | 0.31 | 0.31 | 0.31 | 0.30 | 0.32 | 0.38 | 0.34 |
| Yb | 2.14 | 2.05 | 1.91 | 1.95 | 1.79 | 2.34 | 2.21 | 1.97 | 1.91 | 1.89 | 2.09 | 2.00 | 2.00 | 2.02 | 2.11 | 2.27 | 2.19 | 2.19 | 2.16 | 2.42 | 2.14 |
| Lu | 0.31 | 0.29 | 0.27 | 0.29 | 0.26 | 0.41 | 0.38 | 0.33 | 0.32 | 0.32 | 0.32 | 0.34 | 0.32 | 0.35 | 0.36 | 0.39 | 0.38 | 0.38 | 0.35 | 0.40 | 0.32 |
| Y | 19.79 | 18.48 | 17.48 | 17.29 | 16.25 | 18.20 | 16.95 | 16.12 | 15.30 | 15.59 | 19.44 | 17.08 | 19.44 | 19.45 | 19.14 | 16.77 | 16.63 | 16.01 | 18.59 | 22.62 | 20.46 |
| ΣREE | 281.96 | 243.03 | 242.72 | 250.69 | 239.80 | 103.52 | 125.07 | 124.25 | 138.87 | 125.75 | 255.04 | 110.65 | 119.69 | 98.04 | 104.77 | 99.60 | 98.54 | 93.43 | 101.44 | 100.41 | 98.76 |
| LREE | 265.49 | 227.89 | 228.52 | 236.24 | 226.24 | 90.65 | 112.72 | 112.49 | 126.89 | 113.67 | 238.99 | 98.56 | 107.48 | 85.44 | 91.62 | 87.92 | 87.12 | 82.19 | 87.84 | 83.49 | 83.16 |
| HREE | 16.47 | 15.14 | 14.20 | 14.44 | 13.56 | 12.87 | 12.35 | 11.75 | 11.98 | 12.08 | 16.05 | 12.10 | 12.20 | 12.59 | 13.15 | 11.69 | 11.41 | 11.24 | 13.61 | 16.91 | 15.60 |
| LREE/HREE | 16.12 | 15.05 | 16.09 | 16.36 | 16.68 | 7.04 | 9.13 | 9.57 | 10.59 | 9.41 | 14.89 | 8.15 | 8.81 | 6.79 | 6.97 | 7.52 | 7.63 | 7.31 | 6.45 | 4.94 | 5.33 |
| La$_N$/Yb$_N$ | 23.67 | 20.92 | 23.25 | 23.55 | 24.38 | 5.72 | 8.42 | 9.90 | 11.62 | 10.54 | 21.19 | 7.91 | 8.87 | 6.24 | 6.56 | 6.05 | 6.22 | 5.80 | 5.78 | 4.08 | 4.84 |
| δEu | 0.60 | 0.65 | 0.70 | 0.70 | 0.72 | 0.79 | 0.73 | 0.73 | 0.74 | 0.72 | 0.63 | 0.83 | 0.74 | 0.67 | 0.65 | 0.83 | 0.85 | 0.85 | 0.77 | 0.71 | 0.87 |
| δCe | 0.97 | 0.97 | 0.97 | 0.97 | 0.97 | 1.00 | 0.99 | 0.98 | 0.98 | 0.96 | 0.97 | 0.97 | 0.96 | 0.96 | 0.97 | 0.97 | 0.98 | 0.98 | 0.95 | 0.98 | 0.96 |

Note: A/NK = $Al_2O_3$/($Na_2O$ + $K_2O$) (mol); A/CNK = $Al_2O_3$/(CaO + $Na_2O$ + $K_2O$) (mol); Mg$^{\#}$ = 100 × Mg/(Mg + total Fe) [48].

**Table 5.** Rb–Sr and Sm–Nd isotopic data for the host Sanguliu granodiorite and monzodiorite enclaves.

| Sample | Rb (ppm) | Sr (ppm) | $^{87}$Rb/$^{86}$Sr | $^{87}$Sr/$^{86}$Sr | I$_{Sr}$ | $\varepsilon_{Sr}$(0) | $\varepsilon_{Sr}$(t) | f$_{Rb/Sr}$ | Sm (ppm) | Nd (ppm) | $^{147}$Sm/$^{144}$Nd | $^{143}$Nd/$^{144}$Nd | I$_{Nd}$ | T$_{DM}$ | T$_{2DM}$ | T$_{chur}$ | $\varepsilon_{Nd}$(0) | $\varepsilon_{Nd}$(t) | f$_{Sm/Nd}$ |
|---|---|---|---|---|---|---|---|---|---|---|---|---|---|---|---|---|---|---|---|
| SGL−1 | 149 | 383 | 1.1227 | 0.716629 | 0.71467 | 172.2 | 146.4 | 12.58 | 6.65 | 42.8 | 0.094 | 0.511579 | 0.511503 | 1997 | 2467 | 1569 | −20.66 | −19.06 | −0.52 |
| SGL-2 | 166 | 389 | 1.2341 | 0.716771 | 0.71461 | 174.2 | 145.6 | 13.92 | 6.36 | 43.1 | 0.0893 | 0.511577 | 0.511505 | 1924 | 2465 | 1503 | −20.7 | −19.02 | −0.55 |
| SGL-3 | 176 | 369 | 1.377 | 0.717103 | 0.7147 | 178.9 | 146.9 | 15.65 | 5.81 | 38.3 | 0.0917 | 0.511584 | 0.51151 | 1953 | 2456 | 1527 | −20.56 | −18.92 | −0.53 |
| SGL-4 | 189 | 378 | 1.4503 | 0.717021 | 0.71449 | 177.7 | 143.9 | 16.54 | 6.21 | 42.2 | 0.0889 | 0.511559 | 0.511488 | 1940 | 2493 | 1523 | −21.05 | −19.35 | −0.55 |
| SGL-5 | 177 | 395 | 1.2954 | 0.716622 | 0.71436 | 172.1 | 142 | 14.66 | 6.06 | 40.7 | 0.09 | 0.511458 | 0.511386 | 2080 | 2653 | 1682 | −23.02 | −21.34 | −0.54 |
| BT-1 | 199 | 291 | 1.9851 | 0.718026 | 0.7145 | 192 | 144.1 | 23 | 3.91 | 19.4 | 0.122 | 0.511598 | 0.511498 | 2571 | 2469 | 2114 | −20.29 | −19.11 | −0.38 |
| BT-2 | 178 | 303 | 1.7026 | 0.717685 | 0.71466 | 187.2 | 146.3 | 19.59 | 4.15 | 22.5 | 0.1114 | 0.511523 | 0.511432 | 2417 | 2575 | 1986 | −21.75 | −20.39 | −0.43 |
| BT-3 | 176 | 321 | 1.5825 | 0.717302 | 0.71449 | 181.7 | 143.9 | 18.14 | 3.86 | 20.4 | 0.1143 | 0.510985 | 0.510892 | 3300 | 3423 | 3037 | −32.24 | −30.93 | −0.42 |
| BT-4 | 175 | 327 | 1.546 | 0.717299 | 0.71455 | 181.7 | 144.8 | 17.69 | 4.08 | 22.1 | 0.1117 | 0.511624 | 0.511533 | 2274 | 2416 | 1813 | −19.78 | −18.42 | −0.43 |
| BT-5 | 170 | 329 | 1.4977 | 0.717262 | 0.7146 | 181.2 | 145.5 | 17.11 | 4.11 | 21.5 | 0.1159 | 0.511443 | 0.511348 | 2650 | 2705 | 2245 | −23.31 | −22.03 | −0.41 |

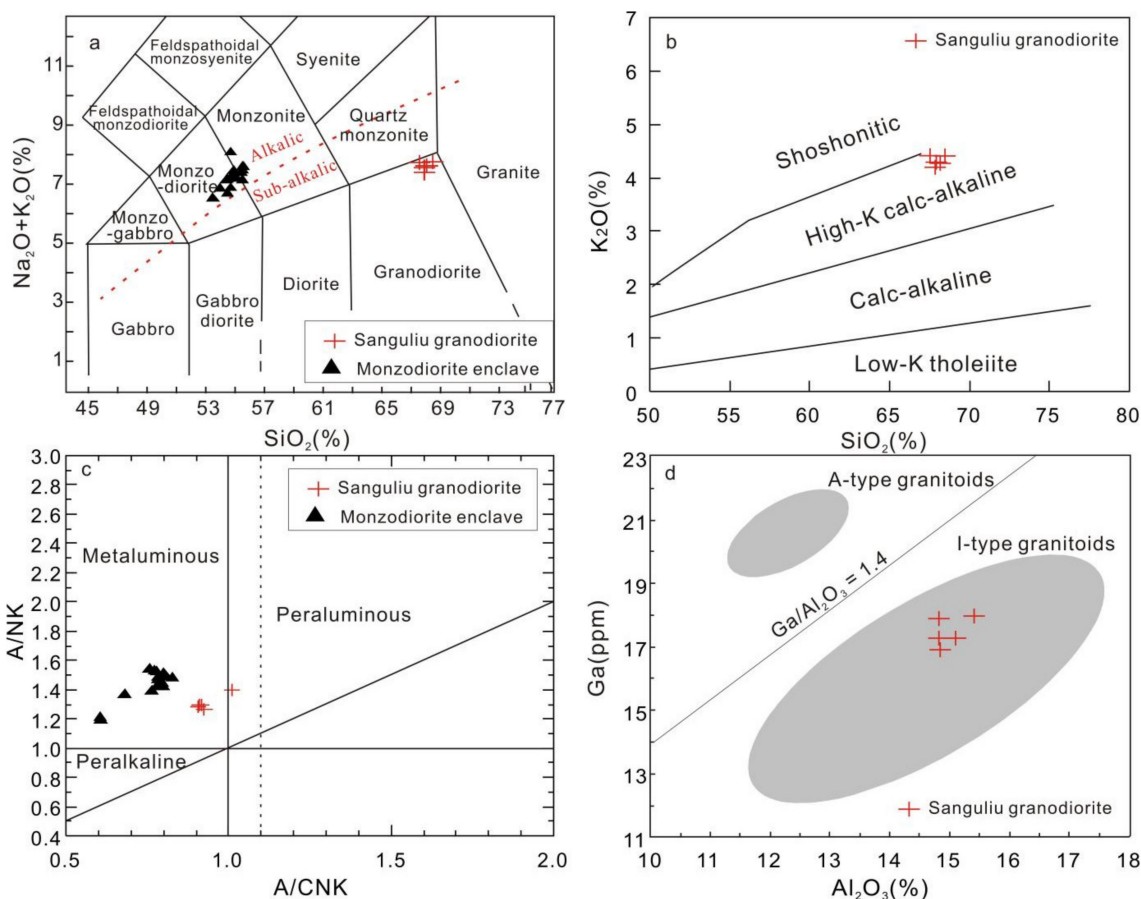

**Figure 8.** TAS (**a**), SiO$_2$ versus K$_2$O (**b**), A/CNK versus A/NK (**c**), and Al$_2$O$_3$ versus Ga (**d**) diagrams of the Sanguliu granodiorite and its enclaves.

The Sanguliu granodiorite samples have medium total REE contents ($\sum$REE), ranging from 239.80 to 281.96 ppm, and exhibit strong differentiation between the light rare earth element (LREE) and heavy rare earth element (HREE), demonstrating LREE/HREE and La$_N$/Yb$_N$ ratios of 15.05–16.68 and 20.92–24.38, respectively. Further, their REE patterns present an LREE-enriched pattern with medium negative Eu anomalies ($\delta$Eu = 0.60–0.72) (Figure 9a). In comparison, the monzodioritic enclave samples have lower REE contents ($\sum$REE = 93.43–138.87 ppm), a slightly flatter LREE segments pattern, and LREE/HREE amd La$_N$/Yb$_N$ ratios of 4.94–10.59 and 4.08–11.62, respectively. In addition, they feature weak negative Eu anomalies ($\delta$Eu = 0.65–0.85) (Figure 9a).

The granodiorite and monzodioritic enclave samples yielded similar variations of trace elemental compositions in the primitive mantle-normalized trace element spider diagram (Figure 9b). They are characterized by enrichments in large ion lithisphile elements (LILEs, i.e., Rb and U), high field strength elements (HFSEs, i.e., Zr, Hf, and Ti), and depletions in LILEs (i.e., K and Sr) and HFSEs (i.e., Nb, P, and HREE) (Figure 9b).

### 5.4. Sr–Nd–Pb Isotopes

The Rb–Sr and Sm–Nd isotope data for the host granodiorite and monzodioritic enclave samples are listed in Table 5 and shown in Figure 10a. The initial $^{87}$Sr/$^{86}$Sr ratios and $\varepsilon_{Nd}(t)$ values were calculated at 122 and 125 Ma from the zircon U–Pb dating. All the samples exhibited similar Sr isotopic features with high $^{87}$Rb/$^{86}$Sr (1.1227 to 1.9851) and initial $^{87}$Sr/$^{86}$Sr ratios (0.71436 to 0.71470). In addition, both the host granodiorite and monzodiorite enclaves are characterized by strongly negative $\varepsilon_{Nd}(t)$ values of −18.92−−21.34, and

$-18.42$-$-30.93$ (mostly $-18.42$-$-22.03$), respectively. Furthermore, the enclave samples were scattered in the $\varepsilon_{Nd}(t)$–SiO$_2$ diagram (Figure 10b).

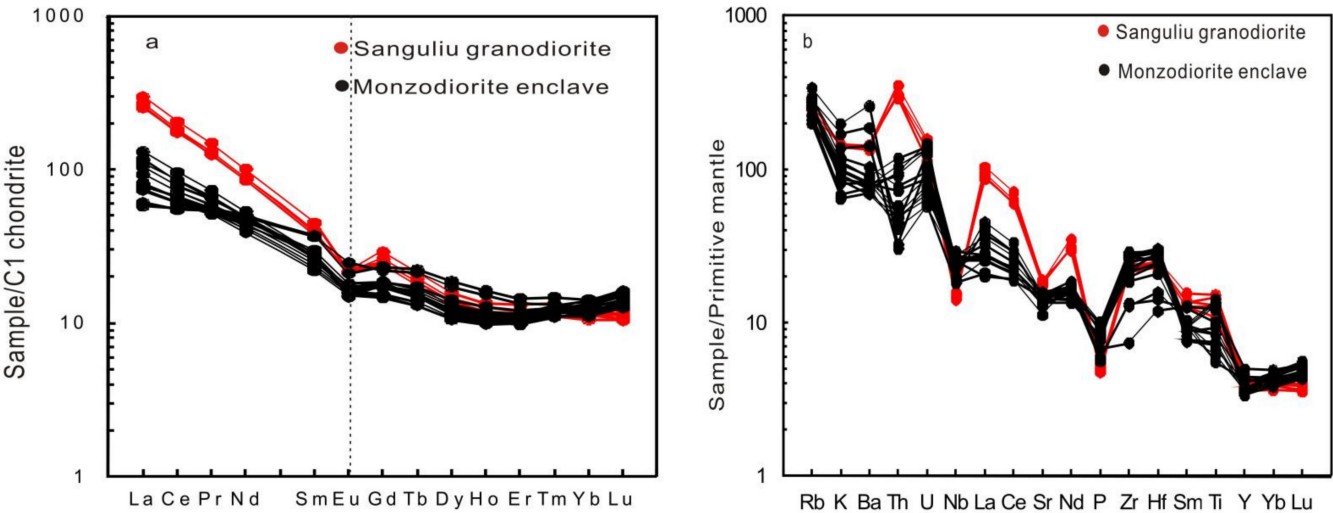

**Figure 9.** Chondrite-normalized REE patterns (**a**) and primitive mantle-normalized spiderdiagrams (**b**) for the Sanguliu granodiorite and monzodioritic enclaves. The chondrite and primitive mantle values are from [49].

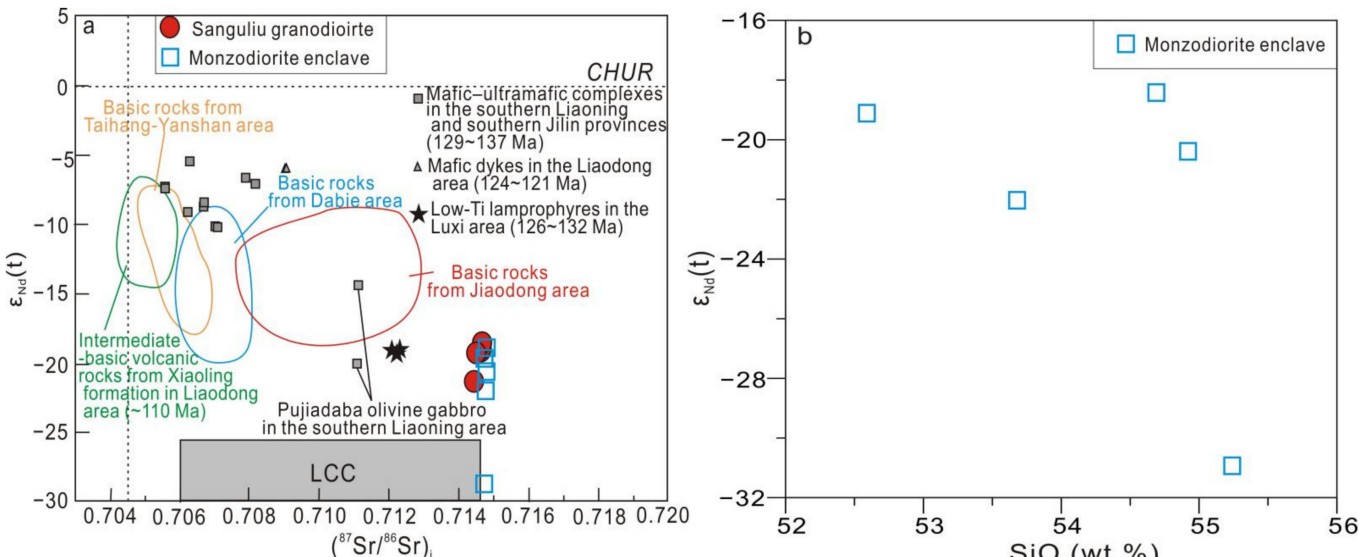

**Figure 10.** $\varepsilon_{Nd}(t)$ versus ($^{87}Sr/^{86}Sr)_i$ plot (**a**) and SiO$_2$ versus $\varepsilon_{Nd}(t)$ diagram (**b**). The data of the mafic rocks in the Taihang-Yanshan area are sourced from [50,51]; those in the Dabie area are from [52–54]; those in the Jiaodong area are from [55–57]; those in the Southern Jilin and Southern Liaoning areas are from [58]; the intermediate-basic rocks from the Liaodong area are from [59,60]; lamprophyres in the Luxi area are from [61]; LCC–lower crust of the NCC, which was taken from [62].

Moreover, the host granodiorite and monzodioritic enclave generated similar Pb isotopic ratios: ($^{206}Pb/^{204}Pb)_i$ = 17.6200–17.8360, ($^{207}Pb/^{204}Pb)_i$ = 15.5970–15.6190, and ($^{208}Pb/^{204}Pb)_i$ = 38.8410–39.9490, as summarized in Table 6 and Figure 11.

**Table 6.** Pb isotopic data for the host Sanguliu granodiorite and monzodiorite enclaves.

| Sample | $^{206}Pb/^{204}Pb$ | $^{207}Pb/^{204}Pb$ | $^{208}Pb/^{204}Pb$ | $(^{206}Pb/^{204}Pb)_i$ | $(^{207}Pb/^{204}Pb)_i$ | $(^{208}Pb/^{204}Pb)_i$ |
|---|---|---|---|---|---|---|
| SGL-1 | 17.914 | 15.627 | 39.484 | 17.746 | 15.619 | 38.984 |
| SGL-2 | 17.955 | 15.615 | 39.338 | 17.82 | 15.608 | 38.933 |
| SGL-3 | 17.838 | 15.609 | 39.271 | 17.697 | 15.602 | 38.841 |
| SGL-4 | 17.989 | 15.607 | 39.57 | 17.836 | 15.6 | 39.153 |
| SGL-5 | 17.847 | 15.616 | 39.276 | 17.738 | 15.611 | 38.901 |
| BT-1 | 17.836 | 15.613 | 38.987 | 17.732 | 15.608 | 38.877 |
| BT-2 | 17.83 | 15.609 | 39.133 | 17.648 | 15.6 | 38.949 |
| BT-3 | 17.82 | 15.617 | 39.138 | 17.681 | 15.61 | 39.011 |
| BT-4 | 17.799 | 15.615 | 39.09 | 17.654 | 15.608 | 38.929 |
| BT-5 | 17.78 | 15.605 | 38.995 | 17.62 | 15.597 | 38.858 |

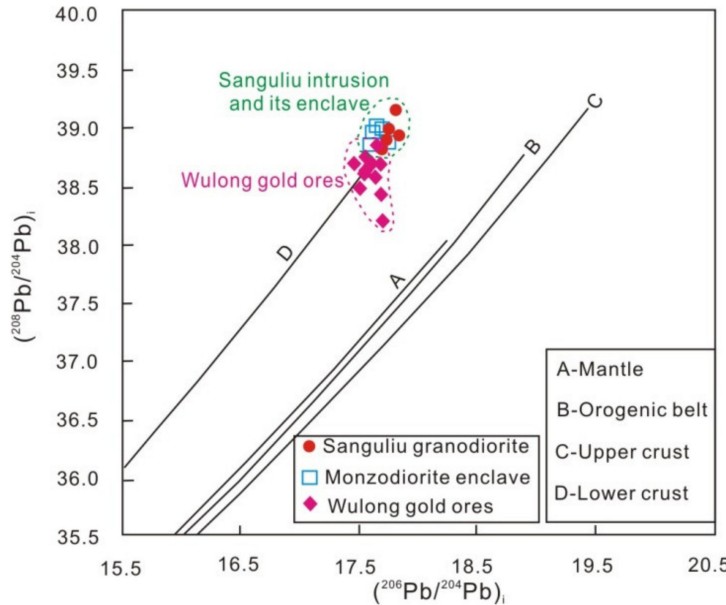

**Figure 11.** $(^{206}Pb/^{204}Pb)_i$ versus $(^{208}Pb/^{204}Pb)_i$ diagram (modified from [63]). The data of the Wulong gold deposit are from [17,64].

### 5.5. Biotite Compositions

Six biotite mineral spots on the host Sanguliu granodiorite were analyzed, and their EPMA data are listed in Table 7. The cations were initially calculated based on 22 oxygen atoms [65]. The EPMA data indicate that the biotite has high $TiO_2$ (2.71 to 3.71 wt.%), MgO (10.54 to 11.48 wt.%), FeO (20.10 to 21.49 wt.%), and $K_2O$ (9.24 to 9.87 wt.%) contents, and low CaO (0.01 to 0.024 wt.%) and $Na_2O$ (0.06 to 0.11 wt.%) contents. Further, the calculated $Fe^{2+}/(Fe^{2+} + Mg)$, $Mg/(Mg + Fe^{2+})$ ratios, and $Al^{VI}$ values were found to be 0.46–0.50, 0.47–0.50, and 0.058–0.112, respectively. In the $Mg-Al^{VI} + Fe^{3+} + Ti-Fe^{2+} + Mn$ diagram, the biotite from the host Sanguliu granodiorite was classified as magnesio-biotite (Figure 12a). In the $Fe^{2+}-Fe^{3+}-Mg^{2+}$ diagram, all of the samples plotted between the NNO and MH buffers (Figure 12b).

**Table 7.** Electron microprobe data of the biotites from the host Sanguliu granodiorite (results in wt%).

| Sample Element | 2-bt-1 | 2-bt-3 | 2-bt-5 | 2-bt-6 | 2-bt-7 | 2-bt-8 |
|---|---|---|---|---|---|---|
| $SiO_2$ | 36.453 | 37.332 | 37.391 | 36.849 | 36.95 | 36.949 |
| $TiO_2$ | 3.622 | 3.713 | 2.707 | 3.609 | 3.089 | 2.586 |
| $Al_2O_3$ | 13.694 | 13.502 | 13.83 | 13.483 | 13.581 | 13.406 |
| FeO | 21.493 | 20.458 | 20.099 | 21.147 | 20.759 | 21.292 |
| MnO | 0.246 | 0.248 | 0.232 | 0.258 | 0.281 | 0.255 |
| MgO | 10.554 | 10.947 | 11.478 | 10.758 | 10.547 | 11.1 |
| CaO | 0.011 | 0.014 | 0.013 | 0.001 | 0.024 | 0.016 |
| $Na_2O$ | 0.077 | 0.055 | 0.129 | 0.108 | 0.105 | 0.09 |
| $K_2O$ | 9.399 | 9.865 | 9.513 | 9.348 | 9.239 | 9.344 |
| Si | 2.8127 | 2.8494 | 2.8637 | 2.8350 | 2.8643 | 2.8598 |
| $Al^{IV}$ | 1.1873 | 1.1506 | 1.1363 | 1.1650 | 1.1357 | 1.1402 |
| $Al^{VI}$ | 0.0580 | 0.0639 | 0.1120 | 0.0576 | 0.1051 | 0.0826 |
| Ti | 0.2102 | 0.2132 | 0.1560 | 0.2089 | 0.1801 | 0.1506 |
| $Fe^{3+}$ | 0.1772 | 0.1855 | 0.1695 | 0.1884 | 0.2002 | 0.1537 |
| $Fe^{2+}$ | 1.2097 | 1.1204 | 1.1179 | 1.1722 | 1.1457 | 1.2245 |
| Mn | 0.0161 | 0.0160 | 0.0150 | 0.0168 | 0.0185 | 0.0167 |
| Mg | 1.2140 | 1.2456 | 1.3105 | 1.2339 | 1.2188 | 1.2807 |
| Ca | 0.0009 | 0.0011 | 0.0011 | 0.0001 | 0.0020 | 0.0013 |
| Na | 0.0115 | 0.0081 | 0.0192 | 0.0161 | 0.0158 | 0.0135 |
| K | 0.9252 | 0.9606 | 0.9295 | 0.9175 | 0.9137 | 0.9226 |
| Total | 7.8228 | 7.8145 | 7.8305 | 7.8116 | 7.7998 | 7.8463 |

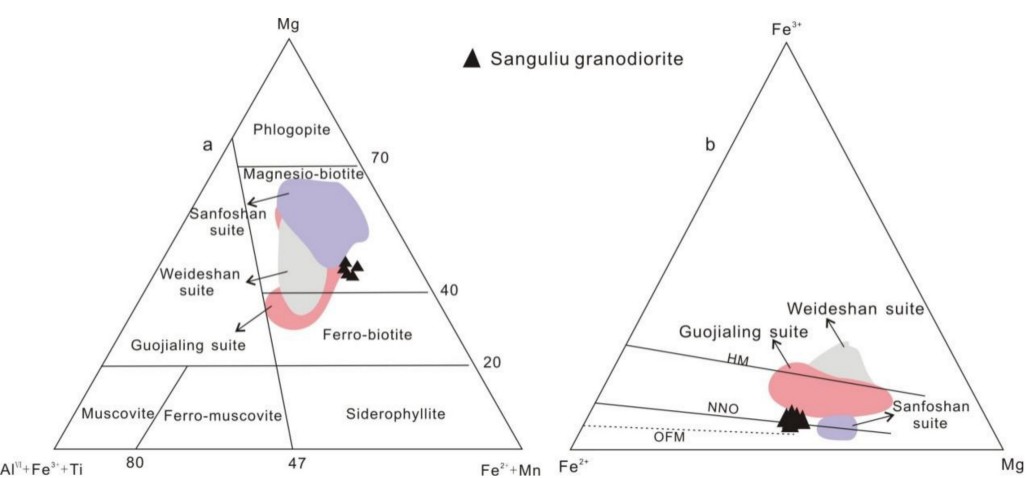

**Figure 12.** Diagrams of classification ((**a**), modified from [66]) and $Fe^{3+}$–$Fe^{2+}$–Mg ((**b**), modified from [67]) of the biotites from the Sanguliu, Guojialing, Weideshan, and Sanfoshan plutons. The data of EPMA biotite compositions are from [23,68] and this paper, respectively).

## 6. Discussion

### 6.1. Petrogenesis

The Sanguliu granodiorite samples show relatively low contents of $SiO_2$ (67.54%–68.45%) and $Al_2O_3$ (14.82%–15.39%), and low A/CNK values (0.92–1.02), indicating that Sanguliu granodiorite belongs to calc-alkaline I-type granite [69,70]. In addition, the granodiorite samples also exhibit low Ga/Al ratios (Figure 8d), similar to I-type granites, which are different from A-type granite (Figure 8d) [71]. This also could be supported by the petrographic features of the Sanguliu granodiorite. The Sanguliu granodiorite consists of quartz, plagioclase, potassium feldspar, biotite, and minor hornblende (Figure 4), with no muscovite. The petrographic and geochemical data indicate that the Sanguliu granodiorite belongs to typical calc-alkaline I-type granite.

The monzodioritic enclaves found in the host Sanguliu intrusion via field investigation may provide significant insights into the magma processes present [2]. The enclaves in the granitoids are generally interpreted as restites, country-rock xenoliths, autoliths, schlieren, and MMEs [2,72]. The monzodioritic enclaves in the Sanguliu intrusion are usually oval or elliptical and have typical igneous microtextures and sharp contacts with the host rocks. The monzodioritic enclaves have homogeneous U-Pb ages (125.3 ± 1.6 Ma, SHRIMP) with the host Sanguliu intrusion (122.2 ± 1.3 Ma, LA-ICP-MS), which are within the range of acceptable error. Furthermore, the monzodioritic enclaves yield similar medium to weakly negative Eu anomalies and relative Sr depletion to that of the host intrusion, as shown in Figure 9, thereby excluding the possibility of an autolith characterization [73]. These petrographic, geochemical, and geochronological features indicate that the monzodioritic enclaves in the Sanguliu intrusion were not restitic, xenoliths of wallrock, autoliths, or schlierens, but rather MMEs in origin.

MMEs usually represent remnants of a mafic component added to intermediate or felsic magma chambers [4,8,72,74,75]. The MMEs in the Sanguliu intrusion generally contain acicular apatites and megacrysts. The presence of both acicular apatites, which are attributed to rapid cooling, and megacrysts, such as K-feldspar or plagioclase, which are elliptical, implies that the enclaves are hybrids, meaning that they are related to the mingling of small volumes of hot basalt with a cooler granitic melt [3,76]. Furthermore, the MMEs are characterized by wide ranges of $\varepsilon_{Nd}(t)$ values ($-18.42$–$-30.93$), which have up to 10 epsilon units. These features suggest that magma mixing between mafic and granitic melts played an important role in the genesis of the Sanguliu granodiorite and MMEs.

The MMEs and host Sanguliu granodiorite show similar whole-rock Sr and isotopic compositions (Figure 11), indicating rapid homogenization and radiogenic isotopic equilibrium during the magma mixing process [24,77–79]. In addition to magma mixing, crystal fractionation was involved in the formation of the Sanguliu granodiorite, which is supported by the weak to medium negative Eu anomalies (Figure 9a) and the significant Sr and P depletion shown in the spidergram (Figure 9b). Therefore, the Sanguliu granodiorite and MMEs were the result of mixing between mafic and intermediate to felsic magmas, followed by apatite- and feldspar-dominated fractionation. The magma mixing process likely occurred within the primary magma chambers rather than during magma ascent or evolution, which is reflected by the lack of correlations between the $\varepsilon_{Nd}(t)$ and $SiO_2$ contents of the enclaves (Figure 10b) [80].

*6.2. Magma Source*

The host granodiorite has high $SiO_2$ (67.54 to 68.45 wt.%) and low MgO (1.51 to 1.72 wt.%), $TiO_2$ (0.47 to 0.52 wt.%) contents, and strongly negative $\varepsilon_{Nd}(t)$ values of $-15.53$ to $-21.34$, and $\varepsilon_{Hf}(t)$ values of $-20.86$ to $-25.31$, indicating a crustal derivation. The primitive magma can be inferred to have isotopic compositions with $\varepsilon_{Hf}(t)$ values less than $-25.31$ and ancient Hf model ages ($T_{DM2}$, 2501–2781 Ma, [45]). As the samples plotted close to the ancient lower crust of the NCC field in the t-$\varepsilon_{Hf}(t)$ diagram (Figure 7a), the host granodiorite may have been primarily derived from an ancient lower crust source. However, the high $Mg^{\#}$ values of 43.73–47.40 of the host granodiorite suggest the involvement of mantle-derived material as a result of magma mixing.

The MMEs have high MgO (4.18 to 6.17 wt.%), Cr (45.91 to 290.04 ppm), and Ni (19.65 to 88.18 ppm) contents, and high $Mg^{\#}$ values of 50–57 at intermediate silica contents ($SiO_2$ = 53.68–55.78 wt.%), making them distinct from crustal materials or crust-derived melts [81], thereby suggesting that the parental magma of the enclaves was mantle-derived basaltic magma. However, the MMEs yielded strongly negative $\varepsilon_{Nd}(t)$ values of $-18.42$ to $-30.93$, which may be the result of the partial melting of an enriched lithospheric mantle or a depleted lithospheric mantle with significant contamination by ancient crustal materials during magma evolution [80]. The voluminous incorporation of crustal components significantly modifies the concentrations of major and trace elements and $Mg^{\#}$ values [47,80]. However, the relatively low $SiO_2$ and high MgO contents and $Mg^{\#}$ values indicate that

significant crustal contamination during magma evolution did not occur. Therefore, the enclave-forming magmas were likely derived mainly from an enriched mantle source.

　　The enclaves yielded lower negative $\varepsilon_{Nd}(t)$ values (−18.42 to −30.93) than those of the majority of Early Cretaceous mafic rocks in the NCC, which were derived from an enriched lithospheric mantle modified from crustal or slab melts [45,58], and the enclaves plot between the "primitive" mafic rocks from the NCC and ancient lower crust in the diagrams of t–$\varepsilon_{Hf}(t)$ and $\varepsilon_{Nd}(t)$–($^{87}Sr/^{86}Sr)_i$ (Figures 7a and 10a). These clues suggest that the enclave-forming magmas were contaminated by lower crustal melts. This is supported by the distinct LILE and LREE enrichment and HFSE depletion in the enclaves (Figure 9) [82]. The SHRIMP zircon U-Pb age of 1776 ± 24 Ma for the enclaves also indicates the incorporation of NCC basement materials, which are characterized by late Archean and Paleoproterozoic tectonothermal events [83]. Both Pei et al. [58] and Li et al. [84] proposed that the delaminated NCC lower continental crust materials modified the lithospheric mantle in the Liaodong area. Furthermore, the enclaves have highly negative $\varepsilon_{Nd}(t)$ values (−18.42 to −30.93) that are similar to those of the Early Cretaceous low-Ti lamprophyres in the Luxi area and the Pujiadaba olivine gabbro in the southern Liaoning area (Figure 10a). These low-Ti lamprophyres were derived from the partial melting of a lithospheric mantle modified by crustal melts from both the delaminated NCC lower crust and the broken-off YC slab [45] and the Pujiadaba olivine gabbros in the southern Liaoning area were derived from the partial melting of the lithospheric mantle metasomatized by melts from the broken-off YC slab [58]. Therefore, the enclave-forming magma originated from the partial melting of lithospheric mantle modified by the delaminated NCC lower crustal and broken-off YC slab melts.

　　Based on the aforementioned petrography, major and trace element geochemical, and Sr, Nd, Pb, and zircon in situ Hf isotopic compositions of the MMEs and host granodiorite, a complex, multi-stage process involving magma mixing and crystal fractionation, is proposed for the generation of the Sanguliu intrusion and monzodioric enclaves. The Sanguliu granodiorite was mainly derived from the partial melting of the ancient NCC lower crust, which was mixed with mantle-derived magma. Meanwhile, the enclaves were derived from the partial melting of enriched lithospheric mantle modified by melts from the delaminated NCC lower crust and broken-off YC slab, followed by crystal fractionation. The Sr and Pb isotopic equilibrium and crystal exchange occurred during the mixing of mafic-felsic magma.

### 6.3. Highly Oxidized Magma

　　Highly oxidized magmas can incorporate significant amounts of S [12], and thus serve as an effective medium for transporting Au and other ore-forming materials [10]. The primary biotites from the Sanguliu granodiorite yield homogeneous $Fe^{2+}/(Fe^{2+} + Mg)$ ratios of 0.46–0.50, indicating that the biotites were not modified by later fluids [85]. The biotite has high $Mg/(Mg + Fe^{2+})$ ratios of 0.47–0.50, and plots in the magnesium biotite field on the $Mg-Al^{VI} + Fe^{3+} + Ti-Fe^{2+} + Mn$ diagram (Figure 12a) indicating that the biotite is magnesium biotite. Further, biotites show high $TiO_2$ content (2.71 to 3.71 wt.%) and low $Al^{VI}$ values (0.058 to 0.112) and cluster between the MH and NNO buffers (Figure 12b). Therefore, they likely formed under relatively high temperatures at high oxygen fugacities. This is supported by the high Ce/Ce* (1.30 to 107.18, average = 27.29) ratio of zircons from the Sanguliu intrusion [86], further suggesting that the parental magma of the Sanguliu intrusion has a relatively high oxygen fugacity.

　　The fluid inclusion Rb–Sr isochron age (120 ± 3 Ma; [15]), pyrite Rb–Sr isochron age (119 ± 1 Ma; [32]), and sericite $^{40}Ar/^{39}Ar$ plateau age (122.8 ± 0.8 Ma; [87]) date the Wulong gold mineralization. The Sanguliu granodiorite has a zircon U–Pb age of 122.2 ± 1.3 Ma, representing its emplacement age. The formation age of the Wulong gold deposit is in good agreement with this age. Further, the Sanguliu intrusion hosts the Wulong gold orebodies, which indicates that the Sanguliu intrusion may be related to gold mineralization.

The Sanguliu intrusion has been previously considered to be a metallogenetic intrusion of the Wulong gold deposit [14]. However, Wei et al. [15] proposed that the Sanguliu intrusion did not directly provide any ore-forming fluids or materials for Wulong gold mineralization but had similar Sr and Nd isotopic compositions to the deeper material source of the Wulong deposit. The Pb isotopic compositions of the Sanguliu intrusion are consistent with those of the sulfides from the Wulong deposit, both plotting near the lower crust curve (Figure 11), implying that the Pb in the Wulong gold deposit and Sanguliu intrusion was mainly derived from a similar lower crust source. Some Early Cretaceous intrusions hidden at depths below the Wulong ore field have been identified via geophysical detection and have been verified to provide important ore-forming fluids and materials for the Wulong gold mineralization [16,17]. Therefore, we suggest that the Sanguliu intrusion and the concealed metallogenic intrusion below the Wulong ore field originated from a magma chamber with a relatively high oxygen fugacity, which is favorable for gold mineralization.

### 6.4. Magma Mixing and Gold Mineralization

Abundant MMEs were found within the Early Cretaceous granitic plutons in the eastern NCC. Some of the host granitic plutons have been researched in terms of emplacement ages, petrogenesis, and magma sources [5–7,9,11,23,55,56,68,75,88–90]. Significant similarities were found in the Early Cretaceous plutons (Table 8), including high $K_2O$ and $Fe_2O_3$ contents and $Mg^{\#}$ values [5–7,9,11,23,55,56,68,75,88–90], and the presence of magnesian biotites that were crystallized under high oxygen fugacity conditions [68,88,89].

**Table 8.** Characteristics of the Early Cretaceous granitoids with MMEs in the Jiaodong and Liaodong areas.

| Intrusion Features | Sanguliu Intrusion | Gudaoling Intrusion | Guojialing Intrusion | Sanfoshan Intrusion | Weideshan Intrusion | Yashan Intrusion |
|---|---|---|---|---|---|---|
| Location | Liaodong Peninsula | | | Jiaodong Peninsula | | |
| Occurrence | pluton | pluton | pluton | pluton | pluton | pluton |
| Emplacement age | 122.2 ± 1.3 Ma | 120 ± 1 Ma | 127.9 ± 1.3 Ma | 119–114 Ma | 127–105 Ma | 118 Ma |
| Rock type | granodiorite | monzogranite, and minor granodiorite, quartz diorite | granodiorite | medium-coarse- grained monzogranite, with minor syenogranite and medium-fine grained monzogranite | gneiss quartz monzodiorite, quartz monzonite, and monzogranite | granite |
| Enclave | monzodioritic | dioritic | dioritic | monzodioritic | dioritic | dioritic |
| Enclave shape | oval, irregular | angular to oval | oval, round | oval, round | oval, round, irregular | oval, round |
| Main minerals | quartz (20%–30%), plagioclase (35%–45%), potassium feldspar (15%–25%), biotite (10%–15%), and minor hornblende (~5%) | plagioclase (39%–43%), potassium feldspar (24%–29%), quartz (19%–35%), biotite (5%), and hornblende (1%–3%) | quartz (20%–25%), plagioclase (35%–55%), potassium feldspar (10%–30%), and minor biotite and hornblende (4%–5%) | quartz (20%–25%), plagioclase (35%–55%), potassium feldspar (30%–35%), biotite (5%–10%), and hornblende (1%–3%) | quartz (20%–30%), plagioclase (26%–33%), potassium feldspar (27%–30%), hornblende (8%–9%), and biotite (6%–7%) | quartz, plagioclase, potassium feldspar, hornblende, and biotite |

**Table 8.** *Cont.*

| Intrusion Features | Sanguliu Intrusion | Gudaoling Intrusion | Guojialing Intrusion | Sanfoshan Intrusion | Weideshan Intrusion | Yashan Intrusion |
|---|---|---|---|---|---|---|
| Accessory minerals | titanite, zircon, and apatite | apatite, zircon, titanite, and Fe–Ti oxides | magnetite, titanite, zircon, and apatite | magnetite, titanite, zircon, and apatite | magnetite, titanite, zircon, and apatite | zircon, apatite, magnetite, and titanite |
| Distinctive minerals | hornblende and biotite | biotite and hornblende | hornblende and biotite | biotite and hornblende | biotite and hornblende | hornblende and biotite |
| Relation to country rock | intrusive | intrusive | intrusive | intrusive | intrusive | intrusive |
| Relation to fault | cut by NE-trending faults | cut by NNE-trending faults | cut by NE- and NNE-trending faults | cut by NE- and NNE-trending faults | cut by NE-, NNE-, and NW-trending faults | cut by NE-trending faults |
| Relation to gold deposits | hosts Wulong gold deposit | - | hosts major gold deposits | - | hosts Mo, Cu, Pb-Zn, Ag, as well as minor Au deposits | - |
| ACNK | 0.9–1.0 | 1–1.09 | 0.8–1.1 | 0.93–1.04 | - | 0.89–1.03 |
| $SiO_2$ (wt.%) | 67.54–68.45 | 71.44–72.56 | ~67 | 71.16–76.63 | Mainly 65.29–66.65 | 65.5–68.82 |
| $K_2O$ (wt.%) | 4.20–4.40 | 3.61–4.47 | 3.97–4.29 | 4.34–5.22 | 3.95–4.57 | 3.45–4.53 |
| $Na_2O$ (wt.%) | 3.22–3.46 | 3.39–4.89 | 3.90–3.98 | 3.59–4.40 | 3.73–4.50 | 3.47–4.13 |
| $Mg^{\#}$ | 47–53 | Most 45–72 | 35–64 | - | - | - |
| $Fe^{2+}/(Fe^{2+} + Mg)$ ratios of biotite | 44–47 | - | 41.3 | 35–40 | 26–49 | - |
| $\varepsilon_{Nd}(t)$ | granodiorite:−18.92–−21.34; enclave: −18.42–−30.93 (mostly −18.42–−22.03) | monzogranite: −18.5–−20.9; enclave: ~−7.2 | granodiorite: −11.2–−17.5 | monzogranite: −18.5–−17.5; enclave: −15.1 | - | granite: −17.0–−18.2; enclave: −15.9–−17.7 |
| Origin | I type | I type | I type | I type | I type | I type |
| References | This paper | [7,75] | [9,22] | [11] | [6] | [90] |

Zircons from the studied Sanguliu granodiorite and monzodioritic enclaves show typical zoning absorption, euhedral to subhedral shapes, and have high Th/U ratios of 0.78–1.47 and 0.47–1.79, respectively, indicating magmatic origins. Moreover, zircon U-Pb age data indicate that the Sanguliu granodiorite had coeval $^{206}Pb/^{238}U$ ages (122.2 ± 1.3 Ma, LA-ICP-MS) with the monzodiorite enclave (125.3 ± 1.6 Ma, SHRIMP) within the range of acceptable error, suggesting that the Sanguliu intrusion formed in the Early Cretaceous (125–122 Ma). This is consistent with the ages of Gudaoling monzogranite and dioritic enclaves in the Liaodong area (120 ± 1 Ma; [8]), the granitic intrusions containing enclaves in the Jiaodong area (105–130 Ma; [6,11,21,22]), and the Taihang plutons in the Taihang orogeny (127–129 Ma; [20]). Thus, the enclaves hosted within the Early Cretaceous granitic plutons were formed by magma mixing and mingling [10], with significant participation of enriched lithospheric mantle-derived magma [7]. These Early Cretaceous granitic plutons and abundant coeval enclaves indicate that magma mixing was common in the eastern NCC, even in the Taihang orogeny and mainly occurred during 110–130 Ma (Figure 1), which is consistent with the formation ages of the decratonic gold deposits (130–120 Ma) in the Jiaodong and Liaodong areas [91]. Furthermore, some gold deposits or occurrences were located near or hosted within the Early Cretaceous granitic plutons containing MMEs in the eastern NCC (Figure 3a) and are considered to be closely related to the plutons. In

particular, the Guojialing granodiorite is believed to be associated with the Wang'ershan gold deposit in the Jiaodong area [92]. Although the studied Sanguliu granodiorite did not serve as a metallogenic intrusion of the Wulong gold deposit in the Liaodong area, the pluton shares a similar material source with the Wulong gold mineralization. These findings imply that magma mixing is temporally, spatially, and genetically related to the Early Cretaceous gold systems in the eastern NCC.

As previously mentioned, the enclaves within the Sanguliu granodiorite were formed by the partial melting of the mixed magma of the enriched lithospheric mantle and NCC lower crust, and the enriched lithospheric mantle was previously metasomatized by melts derived from the delaminated ancient lower crust and broken-off Yangtze slab. Zhu et al. [93] proposed that the decratonic gold deposits are closely related to the interaction between the lower crust and melts derived from the enriched lithospheric mantle, which could induce the formation of voluminous gold-bearing magma, rapidly creating giant gold accumulations. Li and Santosh [10] summarized the mineralogical, geochemical, and isotopic data of the igneous rocks, ore-forming fluids, and ore minerals from the gold deposits in the NCC and suggested that mantle fluids and metals substantially contributed to the decratonic gold systems. Therefore, the mingling and mixing of mafic magma with felsic magma not only provides insights into the magma chamber process and dynamics [3] but is also a critical process for the involvement of gold-bearing mantle components in magmatic-metallogenic systems [10,12].

During the Early Cretaceous, the Liaodong area was affected by the westward subduction of the Paleo-Pacific Plate and lithospheric thinning in the NCC [10,94], which led to magma mixing and associated large-scale gold mineralization in the NCC [95–97]. A new diagenetic and metallogenic model for the Wulong gold deposit is shown in Figure 13.

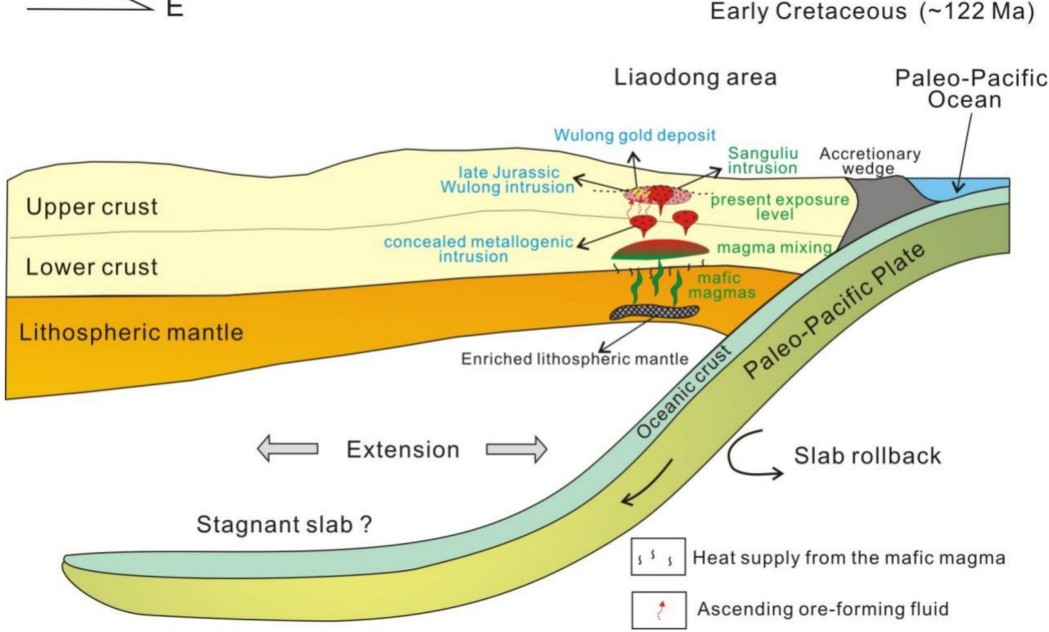

**Figure 13.** Schematic model for the Wulong gold mineralization and its links with the Early Cretaceous magma mixing in the Liaodong Peninsula. The tectonic setting for the Liaodong Peninsula during the Early Cretaceous is from [10,94,97].

## 7. Conclusions

The host Sanguliu granodiorite was emplaced at 122.2 ± 1.3 Ma and showed consistent zircon U-Pb ages with the monzodioritic enclaves (125.3 ± 1.6 Ma), as indicated by LA–ICP–MS and SHRIMP zircon U–Pb age data.

The field and petrological observations, as well as elemental, isotopic, and zircon U-Pb age data, indicate that the enclaves within the Sanguliu intrusion were mainly formed by

magma mixing between mafic and felsic magmas. The felsic magmas were mainly derived from the partial melting of the ancient lower crust, and the mafic magmas were derived from the partial melting of the enriched lithospheric mantle modified by the delaminated ancient lower crust and broken-off Yangtze slab.

The primary magma forming the Sanguliu granodiorite had a relatively high oxygen fugacity, which is favorable for gold mineralization.

In general, the Early Cretaceous magma mixing occurred in the eastern NCC from 110 to 130 Ma and is temporally, spatially, and genetically related to the decratonic gold deposits in the Jiaodong and Liaodong areas.

**Author Contributions:** Conceptualization, T.W. and C.C.; Formal analysis, C.C.; Funding acquisition, C.C. and Y.Z. (Yongheng Zhou); Investigation, C.C., Y.Z. (Yan Zhao) and C.Z.; Methodology, D.L.; Supervision, C.C.; Visualization, C.Z.; Writing—original draft, T.W. and C.C.; Writing—review and editing, T.W. and C.C. All authors have read and agreed to the published version of the manuscript.

**Funding:** This research was funded by the National Key Research and Development Program of China [grant number 2016YFC0600108], and the project from China Geological Survey [grant numbers DD20160346, DD20190379, DD20221806].

**Acknowledgments:** We appreciate the anonymous reviewers for the critical and constructive comments and suggestions. We also deliver special thanks to Hao Yujie and Liu Xiaohe for their support during the zircon LA–ICP–MS U–Pb dating.

**Conflicts of Interest:** The authors declare no conflict of interest.

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
