# Peer review of "The Early Cretaceous Granitoids and Microgranular Mafic Enclaves of Sanguliu Pluton, the Liaodong Peninsula: Implications for Magma Mixing and Decratonic Gold Mineralization in the Eastern North China Craton"

_minerals, doi:10.3390/min12081004_

Round 1
Reviewer 1 Report
The manuscript deals with the very interesting subject of magmatic rock formation and correlation to Au deposits.
The manuscript is well written, the methodology is well described and the results are well presented.
The authors should state why they chose two different methods for U-Pb dating in zircons (LA-ICP-MS for granodiorite zircons) and (SHRIMP for monzodioritic enclaves).
For more details on the review (and the corresponding comments) please see the attached revised version of the manuscript.
Congratulations for your work!

Reviewer 2 Report
Dear authors
I positively have enjoyed and benefited from reading this attractive monograph.
However, I have attached annotated pdf containing some comments (mainly language polishing).

Round 2
Reviewer 2 Report
Dear authors
Thanks for your response to my comments and corrections.